# Adapting Multi-modal Large Language Model to Concept Drift from Pre-training Onwards

**Xiaoyu Yang, Jie Lu, En Yu**
Australian Artificial Intelligence Institute (AAII),
Faulty of Engineering and Information Technology,
University of Technology Sydney, Australia.
`xiaoyu.yang-3@student.uts.edu.au; {jie.lu,en.yu-1}@uts.edu.au`*

## Abstract

Multi-modal Large Language Models (MLLMs) frequently face challenges from concept drift when dealing with real-world streaming data, wherein distributions change unpredictably. This mainly includes gradual drift due to long-tailed data and sudden drift from Out-Of-Distribution (OOD) data, both of which have increasingly drawn the attention of the research community. While these issues have been extensively studied in the individual domain of vision or language, their impacts on MLLMs in concept drift settings remain largely underexplored. In this paper, we reveal the susceptibility and vulnerability of Vision-Language (VL) models to significant biases arising from gradual drift and sudden drift, particularly in the pre-training. To effectively address these challenges, we propose a unified framework that extends concept drift theory to the multi-modal domain, enhancing the adaptability of the VL model to unpredictable distribution changes. Additionally, a T-distribution based drift adapter is proposed to effectively mitigate the bias induced by the gradual drift, which also facilitates the model in distinguishing sudden distribution changes through explicit distribution modeling. Extensive experiments demonstrate our method enhances the efficiency and accuracy of image-text alignment in the pre-training of VL models, particularly in the concept drift scenario. Moreover, various downstream tasks exhibit significant improvements in our model's ability to adapt to the long-tailed open world. Furthermore, we create a set of multi-modal datasets called OpenMMlo, specifically tailored for the long-tailed open-world setting, to validate our findings. To foster the development of the multi-modal community, we have made both OpenMMlo datasets and our code publicly available at: `https://github.com/XiaoyuYoung/ConceptDriftMLLMs`.

## 1 Introduction

The rapid expansion of data availability has created significant challenges for multi-modal large language models (MLLMs), particularly in addressing concept drift, which predominantly manifests as gradual drift and sudden drift Lu et al. (2019). Among them, tailed drift represents a classic illustration of gradual drift, emerging due to severe data imbalance, where the distributions of long-tail categories evolve because of their intrinsic sparsity and noise. Concurrently, sudden drift is mainly represented by OOD drift, as the model encounters new, previously unseen concepts, resulting in distributional shifts that disrupt its ability to generalize in an open-world context. While the issues of long-tailed recognition and concept drift in open-world settings have been extensively studied in visual models Liu et al. (2022b) and language models Kandpal et al. (2023), their impact on MLLMs, particularly vision-language (VL) models, remains largely unexplored. In this work, we aim to bridge this gap by providing a systematic analysis of how tailed drift and OOD drift affect VL models during both pre-training and fine-tuning phases. Our findings highlight critical vulnerabilities of current VL models in adapting to these challenges, underscoring the need for novel strategies to enhance their robustness in dynamic, open-world environments.

---

*Correspondence to Jie Lu and En Yu

**Pre-training:** As illustrated in Figure 1a, a comparison of the VL model trained on the balanced dataset ImageNet Russakovsky et al. (2015b) and the imbalanced dataset ImageNet-LT Liu et al. (2022b) is conducted. Due to the implicit feature centers of each category, we approximate them by averaging unit image and text features obtained by samples on the test set. To assess the intra-class compactness, the cosine distance between the image feature center and the text feature center from the same category is calculated and expressed as degrees. It is evident that training on the imbalanced dataset leads to a higher degree, indicating worse intra-class compactness brought by the tailed drift. Besides, with the tail drift intensifies, it results in a deterioration of the image-text alignment performance in tailed categories. Beyond the deterioration in tailed categories, the tailed drift also affects the image-text alignment in head categories with abundant training samples, which means that it leads to an overall performance degradation, not just the tailed categories. From the perspective of inter-class separability, we measure the average cosine distance from an image feature center to text feature centers of different categories. Figure 1a depicts that the VL model trained on the imbalanced dataset has lower inter-class separability. What's more, we utilize KNN to extract 100 image and text feature centers of OOD samples to verify the impact of OOD drift on the pre-training of the VL model. Compared to training on the balanced dataset, the VL model trained under an imbalanced scenario is harder to distinguish between ID samples and OOD samples from the open world due to their similar inter-class separability. The undistinguished OOD drift will bias the underlying distribution of the feature space in the VL model, further disturbing the image-text alignment in the pre-training.

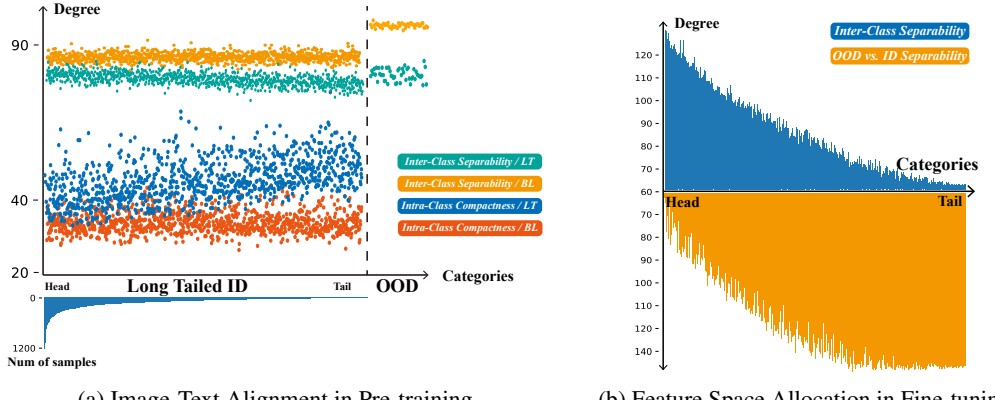

(a) Image-Text Alignment in Pre-training      (b) Feature Space Allocation in Fine-tuning

Figure 1: The impacts of tailed drift and OOD drift on the vision language model in the stages of pre-training and fine-tuning, respectively. (a) In terms of the pre-training, we visualize the alignment results pre-trained on both a balanced dataset (denoted as *BL*) and an imbalanced dataset (as *LT*) without OOD samples, under the same balanced test set. The cosine metric is used to measure the distances between unit image and text features across various categories including OOD samples, which is expressed as degrees. A smaller degree indicates a higher level of similarity between the features. Thus, it provides a feature-level visualization of the intra-class compactness and inter-class separability in the vision language model. (b) In the context of fine-tuning in imbalance datasets, the mutual cosine distance between the centers of each category in the classifier is directly visualized to illustrate the feature space of the classifier, denoted as blue bars. Besides, the average cosine distance between each category center and OOD samples is calculated, which is represented as orange bars.

**Fine-tuning:** We explicitly leverage the weights of the embedding layer in the VL model to visualize its feature space. The average cosine distance between each category and others is calculated as exhibited in the blue bars of Figure 1b. With the decreasing of the training samples, a smaller degree means a worse inter-class separability. It is verified that tail drift leads to a compression of the feature space for tailed categories with a limited number of training samples, while head categories with abundant samples dominate the overall feature space of the VL model. Moreover, the average cosine distance between each category and unit features extracted by OOD samples is applied to reveal the OOD separability as denoted as orange bars in Figure 1b. Since head categories occupy most of the feature space, OOD samples are closer to the center of the head categories compared to the tail categories, implying that in the stage of fine-tuning, it is difficult for the VL model to distinguish between OOD samples under imbalanced scenarios.

To effectively address tailed drift and OOD drift within a unified framework, which often occurs simultaneously, we encapsulate them using the concept drift theory. Therefore, summarizing the above challenges of vision language models in the long-tailed open world, it raises the important question:

*How to adapt multi-modal large language model to concept drift in the long-tailed open world?*

**Remark 1.1. Research Objective:** Our focus lies in addressing the drift that MLLMs exhibit when confronted with the long-tailed open world, rather than leveraging MLLMs to enhance classification performance on long-tailed open datasets. The classification is exploited as the downstream task to provide a more intuitive visualization of the concept drift suffered by the MLLMs, which could have other downstream tasks.

Therefore, we propose a concept drift-aware multi-modal large language model, mitigating the tail drift and OOD drift encountered in the long-tailed open world. Firstly, we introduce the concept drift theory to the multi-modal domain, which provides a more holistic perspective to explain tailed drift and OOD drift. Then, the T-distributed adapter is proposed to be embedded in the hyperspherical feature space. It aligns image-text features for contrastive learning in pre-training. The desirable light-tailed property of the proposed T-distributed spherical metric (Thp) prevents the compression of tailed categories and mitigates the crowding of feature space caused by tailed drift. Besides, in fine-tuning, the adapter projects the features to the decision boundary and detects OOD samples at the feature level based on the underlying distribution. The proposed T-hp distribution explicitly models the feature space with concrete feature centers, optimizing large inter-class margins and yielding more desirable hyperspherical embeddings. And a simple non-parametric KNN is adopted to distinguish the OOD sample based on the T-hp distribution. Finally, we construct a group of multi-modal long-tailed open datasets to support our claims.

In summary, our paper mainly makes the following contributions:

1. We are the pioneers in revealing the unexplored impacts of concept drift to multi-modal large language models, especially in the image-text alignment in the pre-training and feature space allocation in the fine-tuning. This allows future research to more comprehensively study the impact of defect data on MLLMs.

2. The concept drift theory is introduced and extended to multi-modal, integrating the tailed drift and OOD drift in a unified framework. And the T-distributed spherical adapter is proposed to perform the tailed adaptation and OOD detection in the pre-training and fine-tuning stage of the VL model.

3. Extensive experiments evaluate the performance of our method under the long-tailed open world. Compared to specialized models, ours demonstrates superior performance in downstream tasks of long-tailed classification and OOD detection. Crucially, our model effectively addresses drift in image-text alignment, facilitating large-scale pre-training of MLLMs.

4. We build a group of multi-modal datasets OpenMMlo under the long-tailed open world by extending existing image-based long-tailed open datasets. It contains about 740k image-caption pairs with related category annotations. To support and encourage the community focused on multi-modal, we have made both the OpenMMlo and our code public.

## 2 METHODOLOGY

### 2.1 MULTI-MODAL CONCEPT DRIFT THEORY

Concept drift is a phenomenon in which the statistical properties of a target domain change over time in an arbitrary way Lu et al. (2019). Formally, given a set of examples denoted as the data stream $S_{0,t} = \{d_0, ..., d_t\}$, where $d_i = (X_i, y_i)$ is one data instance, $X_i$ and $y_i$ respectively denote the feature vector and the label, and $t$ represents the timestamp of the instance in the data stream. $S_{0,t}$ follows a certain distribution $F_{0,t}(X, y)$. The concept drift is formalized as: $\exists t : P_t(X, y) \neq P_{t+1}(X, y)$, where the joint probability $P_t(X, y)$ can be decomposed as $P_t(X, y) = P_t(X) \times P_t(y|X)$. Although the concept drift due to tailed and OOD data often co-occur, they are fundamentally distinct phenomena. The tailed concept drift foucus on the drift in $P_t(X)$, while $P_t(y|X) = P_{t+1}(y|X)$ remains unchanged. However, the OOD drift from the unknown categories triggers the drift of both $P_t(y|X)$ and $P_t(X)$, that $P_t(y|X) \neq P_{t+1}(y|X)$ and $P_t(X) \neq P_{t+1}(X)$. Therefore, concept drift theory

provides a unified framework to harmonize the tailed shift and OOD shift that often occur together, enabling more robust and adaptive deep learning models.

In the context of multi-modal vision language models, we extend the concept drift theory from a single data stream to multiple data streams. Each modality is associated with a distinct data stream. Thereby, the multi-modal concept drift framework can robustly handle the complex, heterogeneous data distributions inherent to vision language models. Therefore, we formally define multi-modal concept drift as follows:

**Definition 2.1.** *Assume that there are $N$ data streams corresponding to $N$ modalities, given a set of examples denoted as $S_{0,t} = \{S_0, ..., S_i, ..., S_t\}$, where $S_i = (s_1, ..., s_j, ..., s_N)$ and $s_j = (X_{ij}, y_i)$ is one data instance from a single $j$-th data stream, $X_{ij}$ is the feature vector, $y_i$ is the label and $t$ is the timestamp of the data stream. $S_{0,t}$ follows a certain distribution $F_{0,t}(S_i)$, the multi-modal concept drift occurs at timestamp $t + 1$, if $P_{0,t}(S_i) \neq P_{t+1,\infty}(S_i)$, denoted as:*

$$\exists t : P_t(S_i) \neq P_{t+1}(S_i). \tag{1}$$

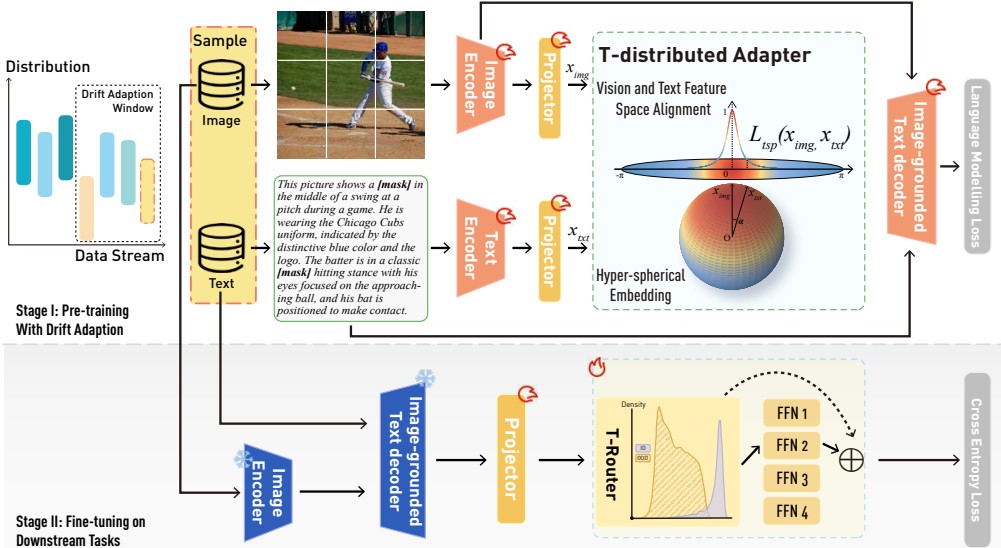

Figure 2: The workflow of our methodology, which consists of two stages: the pre-training of the vision-language model and the fine-tuning on downstream tasks. Within the data streaming, a drift adaptation window slides to detect changes in data distribution and subsequently update the model, in both pre-training and fine-tuning. In the pre-training, the T-distributed adapter aligns visual and textual feature space by image-text contrastive learning, with a large inter-class margin. Coupled with the language model loss, they drive the training of all modules. In the downstream task, the image encoder and the text decoder are frozen out of training, with a linear projector fusing image-text features. Additionally, a mixture of expert modules is leveraged with the T-distributed adapter as the router, allowing it to effectively adapt tail drift and perform OOD drift detection based on the distribution.

## 2.2 T-DISTRIBUTED ADAPTER FOR CONCEPT DRIFT

To adapt the vision language model to concept drift, it is essential to adapt the model to align with the evolving data distribution, which can be formally defined as:

$$\min_{f^{(t)}, f^{(t+1)}, ..., f^{(t+\tau)}} \sum_{i=t}^{t+\tau} L(f^{(i)}(x^{(i)}), y^{(i)}), \tag{2}$$

where $f^{(t)}$ denotes the vision language model trained by the data stream $S_{t-k,t-1}$ from the drift adaption window with the size of $k$. And the model is driven by the target metric $L$ continuously to adapt the drift in a given time period $[t, t + \tau]$. Thus, one prevalent method for detecting and adapting

to concept drift involves designing metrics based on data distributions that can effectively counteract the impacts of sudden and gradual changes within the time window Jiao et al. (2022); Yu et al. (2024). Building upon this thinking, we integrate directional statistics into distribution-based drift detection and adaptation, proposing a T-distributed adapter to alleviate it in the vision language model. Firstly, we provide an overview of directional statistics in the Appendix A.2.1. Then, we will introduce the T-distributed adapter.

Inspired by the T-SNE Van der Maaten & Hinton (2008), we design a T-distribution based metric in hypersphere (T-hp), which follows the density:

$$p_X(x^{(i)}) \propto \frac{2}{\kappa(1 - \mu^T x^{(i)})}, \tag{3}$$

where $x^{(i)} \in \mathbb{S}^{d-1}$ denotes the unit feature vector, $\mu \in \mathbb{S}^{d-1}$ represents the center of category and $\kappa \geq 0$ symbolizes the concentration of the distribution, with higher values indicating a greater concentration around the center $\mu$. Accordingly, we can get the marginal normalizer:

$$N_T(\kappa, d) = \int_{\mathbb{S}^{d-1}} \frac{2}{1 - \kappa \mu^T x^{(i)}} \mathrm{d}x = \frac{1}{\kappa} 2^{\alpha+\beta-1} \frac{\Gamma(\alpha)\Gamma(\beta)}{\Gamma(\alpha+\beta)}, \tag{4}$$

where $\alpha = \frac{d-1}{2}$, $\beta = \frac{d-3}{2}$, and $\Gamma(\cdot)$ represents the gamma function. Combined with Eq. 9 in Appendix A.2.1, the normalizer $N_X(d)$ of density $p_X(x; \mu)$ is:

$$N_X(\kappa, d) = N_T(\kappa, d) \cdot A_{d-2} = \frac{2^{\alpha+\beta} \pi^\beta}{\kappa} \frac{\Gamma(\alpha)}{\Gamma(\alpha+\beta)}. \tag{5}$$

Thus, the probability density function of the proposed Thp distribution is as follows:

$$p(x^{(i)}) = N_X(d)^{-1} \frac{2}{\kappa(1 - \mu^T x^{(i)})}, \quad x^{(i)} \sim \text{Thp}(\mu). \tag{6}$$

The detailed derivation process is provided in Appendix A.2.

In terms of adapting to tailed concept drift, the Thp metric with a large concentration exhibits a light-tailed property, wherein the probability density function exhibits a faster rate of decay as the values increase, relative to the vMF metric, as illustrated in Figure 3. The high kurtosis of Thp is characterized that it yields high confidence only when the feature vector is sufficiently close to the center of the category, thereby minimizing the influence of head category samples on the tail category centers. Formally, $L_{\text{thp}}(\mu, x^{(i)}) = \frac{2}{\kappa(1 - \mu^T x^{(i)}) + \epsilon}$ denotes the T-distributed metric on hypersphere, where $\epsilon$ is a non-zero value setting to 1 to avoid the denominator of 0, and $L_{\text{vmf}}(\mu, x) = \exp(\kappa \mu^T x)$ represents the vMF metric. Given an unit feature vector $x_{\text{head}}$ from the head categories, the gradient of the metric $L$ over the tailed category center $\mu_{\text{tail}}$ is $\frac{\partial L(\mu_{\text{tail}}, x_{\text{head}})}{\partial \mu_{\text{tail}}}$. Due to $\mu_{\text{tail}}^T x_{\text{head}} \in [-1, 1]$, when $\kappa \geqslant 1$, it is readily obtain that $\frac{\partial L_{\text{thp}}}{\partial \mu_{\text{tail}}} < \frac{\partial L_{\text{vmf}}}{\partial \mu_{\text{tail}}}$. Consequently, the light-tailed Thp distribution effectively counteracts the squeezing of tail categories caused by an overwhelming number of head samples, thereby alleviating the bias induced by the tailed concept drift.

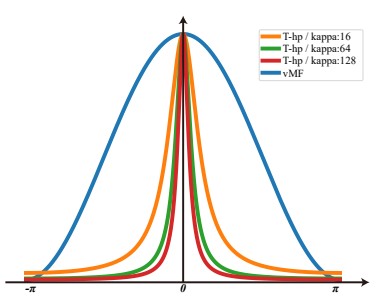

Figure 3: The proposed T-distributed spherical metric with various $\kappa$ and the classical vMF metric when $\kappa = 1$.

Likewise, the Thp metric is directly applied to detect the OOD concept drift. A sample with a unit feature vector $x$ is deemed out-of-distribution if it lies at a relatively large distance from the in-distribution (ID) data in the eigenspace. Following Sun et al. (2022), a simple non-parametric KNN is adopted to partition the data into two sets (ID vs. OOD), which does not impose any distributional assumption on the feature space. Here the distance is the Thp metric with respect to the k-th nearest neighbor.

## 2.3 T-distributed Vision Language Model for the Concept Drift

As illustrated in Figure 2, our proposed vision language model contains two stages, the pre-training and the fine-tuning on the downstream task. Specifically, the classification task is chosen to visualize

the impact of the bias caused by the long-tailed open world on the model. The multi-modal concept drift theory offers a unified framework for integrating gradual drift adaptation and sudden drift detection, where heterogeneous image and text inputs are treated as distinct data streams.

The proposed vision language model follows the encoder-decoder mixture architecture of the Blip Li et al. (2022c), containing an image encoder, a text encoder and an image-grounded text decoder. With an input image $I_i \in \mathbb{R}^{H \times W}$, the visual features $x_{\text{img}}$ are extracted by the image encoder $E_{\text{img}}$ and further projected to the spherical eigenspace by the $L_2$ normalizer $P_{\text{norm}}$:

$$x_{\text{img}} = P_{\text{norm}}(E_{\text{img}}(I_i)) \in \mathbb{R}^{n \times d}, \tag{7}$$

where $n$ is the number of visual features and $d$ represents the feature dimension. The image encoder $E_{\text{img}}$ can be any common visual backbones, such as Vit-Base Dosovitskiy et al. (2021), Vit-Large Dosovitskiy et al. (2021) and ResNeXt-50 Xie et al. (2017). In terms of the text encoder $E_{\text{txt}}$, with a processed input text sequence $T_i$, text features are extracted by the language encoder and further projected to the spherical eigenspace by the $L_2$ normalizer $P_{\text{norm}}$:

$$x_{\text{txt}} = P_{\text{norm}}(E_{\text{txt}}(T_i)) \in \mathbb{R}^{n \times d}, \tag{8}$$

where $n$ is the number of input tokens and $d$ represents the feature dimension. In our case, Bert Devlin et al. (2019a) is used as the language encoder, where a [CLS] token is added to the start of the text input for sentence summarization. Additionally, an image-grounded text decoder is employed to produce a textual description corresponding to a provided image. Utilizing the input visual features $x_{img}$ and text features $x_{txt}$, we initially create fused multi-modal representations by merging the image and text feature embeddings. These combined features act as the keys and values within the cross-attention blocks in the image-grounded text decoder. Through conditioning on the already predicted partial sequence $y_{i<j}$, the decoder iteratively forecasts the token at position $j$, effectively producing textual descriptions corresponding across modalities.

In the pre-training of the vision language model, the T-distributed adapter aligns image and text encoders by contrastive learning. It seeks to align visual and textual transformer feature spaces by promoting similar representations for positive pairs and dissimilar representations for negative pairs. More importantly, our approach circumvents model bias stemming from the long-tailed distribution of data. Specifically, given a mini-batch with $N$ image-text feature pairs, we calculate the $N \times N$ Thp similarity of the cross between image and text features. $N$ correct pairs are recognized as positive samples to maximize the Thp similarity, whereas the rest of $N^2 - N$ are negative samples to minimize the similarity. And we follow the ALBEF Li et al. (2021) to use soft labels from a momentum encoder as training targets to account for the potential positives in the negative pairs.

Additionally, coupled with the T-distributed adapter, language modeling loss is utilized to activate the image-grounding text encoder for generating coherent and detailed captions based on the image, further propelling the training of all three modules. Driven by language modeling loss, the model is trained to optimize a cross-entropy loss with label smoothing, to maximize the likelihood of the generated text in an autoregressive manner.

In the downstream classification task, we leverage the T-distributed adapter as the router to distribute features to various FFNs as experts. Furthermore, by explicitly modeling the feature space, the T-router enables a straightforward application of non-parametric KNN to effectively partition the data into ID and OOD samples. Besides, following the Blip Li et al. (2022c), the image encoder and the text decoder are frozen out of training during fine-tuning. A head of the classifier with two linear layers embeds features into the spherical eigenspace is trained.

## 2.4 BUILDING MULTI-MODAL DATASET OPENMMLO FOR THE LONG-TAILED OPEN WORLD

As the parameters of large models continue to expand, the demand for extensive training data also escalates. However, due to the inherent challenge of obtaining images and related captions, most multi-modal datasets struggle to be balanced in an open world, while cleaning the data requires huge costs. Thus, our aspiration is for the model to adeptly acclimate to the imbalanced dataset by itself, acquiring abundant knowledge with more and more data but not exhibiting bias. In this context, a more realistic training dataset for vision language models is required to validate their potential to be trained under the long-tailed open world. Recognizing the demand for higher-quality multi-modal data with long-tailed distribution in an open world, we developed a group of datasets called Open Multi-modal Long-Tailed OOD Datasets (OpenMMlo).

We extend the open-source datasets, namely ImageNet-LT Liu et al. (2019), iNatualist2018 Van Horn et al. (2018) and Places-LT Liu et al. (2019). ImageNet-LT has 1,000 classes and contains 115.8k samples, with a maximum of 1,280 samples and a minimum of 5 samples for a category. Besides, it consists of 18k images for OOD detection. Places-LT has 184.5K samples from 365 classes, with class samples ranging from 4,980 to 5. The iNaturalist 2018 is a large-scale species dataset collected in the natural world with 437.5K samples for 8,142 classes. We use the InstructBLIP Dai et al. (2023) to generate the related caption of the image, with the prompt of *"What does this picture describe? Please describe in detail its size, location, color, and its relationship to the surroundings."*. And, we define long-tailed data in image-caption pairs according to the image categories, which are provided in open-source image datasets. Concerning related captions based on images, we counted the word frequencies and found that their distribution is similar to the image categories distribution, which is imbalanced. For more details about OpenMMlo, please refer to Appendix A.4.

# 3 EXPERIMENTS

In this section, we first present the performance in downstream long-tailed classification and OOD detection tasks, which is induced by tail drift and OOD drift on MLLMs. Then, we evaluate the interior feature space of the VL model and further demonstrate our method alleviates the crowding and bias problems caused by the tail drift and OOD drift. The constructed long-tailed multi-modal dataset OpenMMlo is utilized for training and validating. In terms of the OOD drift detection, we follow the setting of CIDER Ming et al. (2023). The model is trained on CIFAR100-LT Krizhevsky & Hinton (2009) with an imbalance ratio of 100, and validated on external OOD datasets including SVHN Netzer et al. (2011), Places365 Zhou et al. (2017), LSUN Yu et al., iSUN Xu et al. and Texture Cimpoi et al. (2014). More detailed experimental implementations are given in Appendix A.3.

## 3.1 TAMING THE TAILED DRIFT AND OOD DRIFT FOR ROBUST FINE-TUNING

We compare our proposed vision-language model with other models to explicitly demonstrate its superior performance in long-tailed open-world scenarios. As shown in Table 1, our model demonstrates exceptional overall performance in long-tailed classification across two large-scale datasets, namely the ImageNet-LT and iNaturalist 2018. To ensure a more equitable comparison, we opted to conduct pre-training using ImageNet and iNaturalist datasets separately, rather than pre-training the entire OpenMMlo. Besides, it is worth noting that training from scratch means that our method uses the imbalanced dataset for pre-training instead of utilizing the pre-trained model, such as clip Radford et al. (2021) pre-trained by the large WIT dataset. The results validate the robustness of our vision language model against biases arising from tailed drift, particularly when leveraging large-scale data for both pre-training and fine-tuning.

As shown in Table 1, compared to other methods trained from scratch, especially the ViT model, our model demonstrates a notable lead across all metrics, indicating our effective mitigation of concept drift during the pre-training, and providing robust pre-trained models for downstream tasks. Furthermore, to compare the current long-tailed methods, we apply the same setup as LIFT Shi et al. (2024), i.e., using the pre-trained model of the clip and only fine-tuning. Only fine-tuning means that the method does not pre-train the model on the long-tail dataset, while directly using the parameters of CLIP pre-trained on high-quality and large-scale WIT dataset. And they only fine-tune the model on long-tailed datasets. It is worth noting that, most vision language models only focus on the impact of tailed drift in the fine-tuning, such as LPT DONG et al. (2023), BALLAD Ma et al. (2021), Decoder Wang et al. (2024), VL-LTR Tian et al. (2022) and LIFT Shi et al. (2024). We are the pioneers in revealing the unexplored impacts of concept drift from pre-training onwards. The superior results on medium and few splits of ImageNet-LT demonstrate the adaptability and robustness of our model in dealing with the gradual drift caused by tail data. Besides, although we are slightly behind LIFT Shi et al. (2024) in the few split of iNatualist2018, we still surpass it overall, exhibiting that our method does not compromise the accuracy of the head category to improve the tailed.

Beyond that, we compare our method with the CLIP Radford et al. (2021) under zero-shot, linear probing and fine-tuning, where CLIP results are from the Decoder Wang et al. (2024). Based on the zero-shot results, it is evident that CLIP, even trained on large-scale and high-quality WIT datasets, struggles to address the issue of tailed drift. CLIP only achieves 5.5% accuracy on iNaturalist2018, and the accuracy variance between many-shot and medium-shot scenarios is 11.6% on ImageNet-LT.

Table 1: Evaluation results of long-tailed classification on ImageNet-LT and iNatualist2018. The best-performing models are highlighted in red. Many, Medium and Few denote the evaluated splits of many-shot (>100 training samples), medium-shot (20-100 samples) and few-shot (<20 samples). Top-1 accuracy is applied to evaluate the performance of different methods. Additionally, † means the model is trained with the resolution of $384 \times 384$. Besides, ZS denotes the zero-shot results of the CLIP model, LP represents the linear probing results, and FT means the fine-tuning results.

| Methods | Backbones | ImageNet-LT | | | | iNaturalist 2018 | | | |
|---|---|---|---|---|---|---|---|---|---|
| | | Many | Medium | Few | All | Many | Medium | Few | All |
| **Training from scratch** | | | | | | | | | |
| cRT Kang et al. (2019) | | 61.8 | 46.2 | 27.3 | 49.6 | 69.0 | 66.0 | 63.2 | 65.2 |
| LWS Kang et al. (2019) | | 60.2 | 47.2 | 30.3 | 49.9 | 65.0 | 66.3 | 65.5 | 65.9 |
| MiSLAS Zhong et al. (2021) | | 62.9 | 50.7 | 34.3 | 52.7 | 73.2 | 72.4 | 70.4 | 71.6 |
| BALMS Ren et al. (2020) | | 64.1 | 48.2 | 33.4 | 52.3 | - | - | - | 70.6 |
| LADE Huang et al. (2016) | ResNet-50 | 64.4 | 47.7 | 34.3 | 52.3 | - | - | - | 69.3 |
| ACE Cai et al. (2021) | | - | - | - | 56.6 | - | - | - | 72.9 |
| RIDE Wang et al. (2020) | | 68.2 | 53.8 | 36.0 | 56.9 | 70.9 | 72.4 | 73.1 | 72.6 |
| PaCo Cui et al. (2021) | | 68.2 | 58.7 | 41.0 | 60.0 | 70.3 | 73.2 | 73.6 | 73.2 |
| NCL Li et al. (2022b) | | - | - | - | 57.4 | 72.0 | 74.9 | 73.8 | 74.2 |
| ViT Dosovitskiy et al. (2021) | | 50.5 | 23.5 | 6.9 | 31.6 | 65.4 | 55.3 | 50.9 | 54.6 |
| MAE He et al. (2022) | | 74.7 | 48.2 | 19.4 | 54.5 | 79.6 | 70.8 | 65.0 | 69.4 |
| DeiT Touvron et al. (2022) | | 70.4 | 40.9 | 12.8 | 48.4 | 72.9 | 62.8 | 55.8 | 61.0 |
| LiVT Xu et al. (2023) | ViT-B/16 | 73.6 | 56.4 | 41.0 | 60.9 | 78.9 | 76.5 | 74.8 | 76.1 |
| LiVT † Xu et al. (2023) | | 76.4 | 59.7 | 42.7 | 63.8 | 83.2 | 81.5 | 79.7 | 81.0 |
| **Ours** | | 76.4 | 66.2 | 48.9 | 67.6 | 82.5 | 79.8 | 77.1 | 78.9 |
| **Ours**† | | 77.2 | 68.6 | 51.3 | 69.4 | 83.3 | 82.1 | 80.5 | 81.5 |
| **Only Fine-tuning** | | | | | | | | | |
| CLIP+ZS Radford et al. (2021) | | 82.0 | 70.4 | 69.6 | 70.5 | 9.9 | 5.3 | 4.6 | 5.5 |
| CLIP+LP Radford et al. (2021) | | 87.3 | 65.1 | 19.0 | 67.4 | 62.4 | 7.1 | 0.1 | 10.0 |
| CLIP+FT Radford et al. (2021) | | 83.0 | 65.0 | 39.9 | 68.5 | 79.4 | 67.6 | 59.1 | 65.4 |
| LPT DONG et al. (2023) | ViT-B/16 | - | - | - | - | 62.1 | 76.2 | 79.3 | 76.1 |
| BALLAD Ma et al. (2021) | (w/ Pretrained | 79.1 | 74.5 | 69.8 | 75.7 | - | - | - | - |
| Decoder Wang et al. (2024) | Clip) | - | - | - | 73.2 | - | - | - | 59.2 |
| VL-LTR Tian et al. (2022) | | 84.5 | 74.6 | 59.3 | 77.2 | - | - | - | 81.0 |
| LIFT Shi et al. (2024) | | 80.2 | 76.1 | 71.5 | 77.0 | 72.4 | 79.0 | 81.1 | 79.1 |
| **Ours** | | 79.5 | 76.5 | 74.1 | 77.2 | 83.5 | 82.2 | 80.7 | 81.7 |
| **Ours** † | | 79.9 | 76.8 | 74.5 | 77.6 | 84.1 | 82.7 | 81.0 | 82.1 |

Our method significantly outperforms CLIP in dealing with tailed drift, especially on iNaturalist2018. It also indicates that training on a high-quality balanced dataset alone cannot effectively mitigate the bias induced by long-tail drift. Furthermore, the results of linear probing and fine-tuning demonstrate that imbalanced datasets can induce pronounced tailed drift in MLLMs, ultimately degrading model performance. The CLIP accuracy in Few-shot is only 39.9% under fine-tuning on ImageNet-LT, much lower than the 69.6% under zero-shot. It further verifies the challenges brought by imbalanced data in the training of MLLMs and the superiority of our method in adapting the MLLM to concept drift from pre-training onwards.

In terms of OOD drift detection, our proposed vision language model, trained on CIFAR100-LT as an in-distribution dataset, demonstrates exceptional performance across four diverse OOD datasets, as shown in Table 2. Our approach stands out with two significant advancements. Firstly, the training of our model does not incorporate any additional data from the open world to delineate the decision boundary between ID samples and OOD samples. Secondly, our proposed model detects OOD drift based on the hyperspherical distribution, without the need for any specialized modules. The proposed methodology offers the convenience of training and inference for large models.

Table 2: Evaluation results of OOD detection with the OOD datasets of SVHN, LSUN, iSUN and Texture. ResNet-34 is selected as the image encoder. The best-performing method is highlighted in red. FPR↓ and AUROC↑ are applied to evaluate the performance of different methods.

| Methods | SVHN | | LSUN | | iSUN | | Texture | |
|---|---|---|---|---|---|---|---|---|
| | FPR | AUROC | FPR | AUROC | FPR | AUROC | FPR | AUROC |
| ProxyAnchor Kim et al. (2020) | 87.2 | 82.4 | 37.2 | 91.7 | 70.0 | 85.0 | 65.6 | 85.0 |
| CE + SimCLR Winkens et al. | 24.8 | 94.5 | 56.4 | 89.0 | 66.5 | 83.8 | 63.7 | 82.0 |
| CSI Tack et al. (2020) | 44.5 | 92.7 | 75.6 | 83.8 | 76.6 | 85.0 | 61.6 | 86.5 |
| SSD+ Sehwag et al. (2021) | 31.2 | 94.2 | 79.4 | 85.2 | 80.9 | 84.1 | 66.6 | 86.2 |
| KNN+ Sun et al. (2022) | 39.2 | 92.8 | 49.0 | 89.3 | 75.0 | 82.7 | 57.2 | 88.4 |
| CIDER Ming et al. (2023) | 12.6 | 97.8 | 30.2 | 92.8 | 46.0 | 88.9 | 35.6 | 92.3 |
| Ours | 8.3 | 98.7 | 20.3 | 97.5 | 32.5 | 95.2 | 45.1 | 96.3 |

Moreover, we evaluate the generalizability of our method in the domain generalization setting. Experiments are conducted on ImageNet-Sketch Wang et al. (2019) with ImageNet Russakovsky et al. (2015a) as the source dataset, as shown in the Table 3. From the experiment results, our method achieves superior performance with an accu-

Table 3: Evaluation results of generalization on ImageNet-Sketch Wang et al. (2019) with ImageNet Russakovsky et al. (2015a) as the source dataset. We compare our methods with other VL models, including zero-shot CLIP Radford et al. (2021), linear probing CLIP Radford et al. (2021), CoOp Zhou et al. (2022), VPT Jia et al. (2022) and DAPT Cho et al. (2023).

| CLIP+ZS | CLIP+LP | CoOp | VPT | DAPT | Ours |
|---|---|---|---|---|---|
| 46.1 | 36.0 | 47.1 | 47.7 | 48.3 | 50.2 |

racy of 50.2% on ImageNet-Sketch, attributed to the robustness of the T-distribution-based drift adapter. It further verifies the generalization ability of our model in the open world.

## 3.2 CONCEPT DRIFT-AWARE IMAGE-TEXT ALIGNMENT FOR EFFECTIVE PRE-TRAINING

Moreover, we verified at the feature level that the proposed T-distributed adapter significantly alleviates the bias from tailed drift and OOD drift in the pre-training. As exhibited in Table 4, the degree of ID intra-class compactness reduces from 49.2 in LT/cosine to 36.2 in LT/Thp. It thereby validates the effectiveness of the proposed T-distributed adapter in enhancing feature extraction in long-tailed scenarios. Notably, the decrease in standard deviation demonstrates that the model considerably mitigates the bias induced by tail drift. In addition, our method achieves remarkable inter-class separability within in-distribution categories under long-tailed scenarios, even surpassing the performance of cosine achieved under the balanced dataset. It confirms the effective-

Table 4: Evaluation results of image-text alignment of different contrastive learning strategies in the stage of pre-training, from three perspectives: ID intra-class compactness, ID inter-class separability and the separability between ID and OOD categories. The cosine metric is utilized to measure these distances, which is expressed as average degrees with standard deviation in brackets. We compare our proposed Thp with classical cosine loss, under balanced scenario (BL, ImageNet) and imbalanced scenarios (LT, ImageNet-LT), respectively.

| Pre-training | ID Intra-class Compactness ↓ | ID Inter-class Separability ↑ | ID vs. OOD Separability ↑ |
|---|---|---|---|
| BL / Cosine | 33.0 (±2.85) | 84.3 (±1.64) | 98.4 (±0.98) |
| LT / Cosine | 49.2 (±4.95) | 76.5 (±2.16) | 80.7 (±1.90) |
| LT / Thp | 36.2 (±3.53) | 85.6 (±1.88) | 101.3 (±1.25) |

ness of the proposed high kurtosis method in enhancing the alignment between images and text in the pre-training stage. Concerning the separability between ID and OOD, we achieve superior results even than the balanced condition, attributed to the inherent light-tailed property of the T-distributed adapter. It ensures our approach performs robustly for OOD drift detection.

### 3.3 ABLATION EXPERIMENTS

#### 3.3.1 T-DISTRIBUTED SPHERICAL EMBEDDING IN THE PRE-TRAINING AND FINE-TUNING

Firstly, we conduct ablation experiments to verify the improved performance of the T-distributed adapter in the stage of pre-training and fine-tuning, respectively. As demonstrated in Table 5, our proposed T-distributed adapter exhibits improvements in both the pre-training and fine-tuning stages of the vision language model. It is worth highlighting that the T-distribution adapter plays a more prominent role during the fine-tuning stage. We argue that it is due to different characteristics of the pre-training and fine-tuning. During fine-tuning, the model is directly involved in specific downstream tasks, and explicit category centers are present in the classifier. In contrast, pre-training primarily focuses on aligning image-text features, where the implicit information of categories is embedded. As a result, the T-distribution adapter's impact is more pronounced in the fine-tuning stage compared to pre-training.

Table 5: Ablation evaluation results with or without the T-distributed adapter in the pre-training or fine-tuning. The ✓ denotes the stage is trained with the T-distributed adapter. The results are based on the ImageNet-LT with the Vit-base. Top-1 accuracy (Acc) is used as the metric.

| Pre-training | Fine-tuning | Acc |
|---|---|---|
| - | - | 56.0 |
| ✓ | - | 58.7 |
| - | ✓ | 65.1 |
| ✓ | ✓ | 69.4 |

#### 3.3.2 VARIOUS CONCENTRATION $\kappa$ IN T-ADAPTER

Furthermore, we conduct ablation experiments to examine the impact of concentrations of the T-distributed adapter on the overall performance of the VL models in Table 6. Four fixed degrees are involved, namely $\kappa = 4$, 16, 64 and 128. The greater the degree of concentration, the greater the kurtosis of the Thp metric. Besides, the concentration can also be utilized as a trainable parameter joining in the training, with the initial setting of 16. In Table 6, setting the concentration parameter to $\kappa = 16$ yields superior results. We argue that a smaller concentration makes it challenging to effectively mitigate the biases introduced by tail drift and OOD drift in the vision language model. In terms of the bigger concentration, the model is hard to train due to the high

Table 6: Ablation evaluation results of various concentrations of parameter $\kappa$ on the long-tailed classification task. "Training" denotes the $\kappa$ involved in the training as a parameter of the model with an initial setting of 16. The results are based on the ImageNet-LT with the Vit-base. Top-1 accuracy (Acc) is used as the metric.

| $\kappa$ | 4 | 16 | 64 | 128 | Training |
|---|---|---|---|---|---|
| Acc | 67.2 | 69.4 | 64.9 | 61.1 | 68.3 |

kurtosis of the Thp metric. In the context of concentration as a trainable parameter with the initialization of 16, there is a slight reduction in model performance, accompanied by a marginal increase of the concentration parameter to 16.37. We assert that the introduction of the new parameter increases the model's complexity, thereby making the training process more challenging.

## 4 CONCLUSIONS AND OUTLOOK

Our findings indicate that visual-language models are significantly affected by biases introduced during both pre-training and fine-tuning in long-tailed open-world scenarios. To address this, we propose a concept drift-aware unified framework for visual-language models. This framework incorporates a T-distributed adapter designed to mitigate biases arising from both tailed drift and out-of-distribution (OOD) drift. Additionally, we introduce a comprehensive set of multi-modal datasets (OpenMMlo) tailored to the long-tailed open world, which includes images, captions and related category annotations.

Finally, we hope that our work will inspire future advancements in multi-modal large language models, specifically addressing the mitigation of biases originating from real-world data challenges, such as tailed drift and OOD drift.

ACKNOWLEDGMENT

The work was supported by the Australian Research Council (ARC) under Laureate project FL190100149.

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

# A APPENDIX

## A.1 RELATED WORKS

### A.1.1 MULTI-MODAL LARGE LANGUAGE MODEL

Large Language Models (LLMs) have recently significantly impacted the field of natural language processing. Through alignment techniques such as supervised learning and reinforcement learning with human feedback, LLMs can effectively generalize to perform a wide range of tasks, even with limited training data. A remarkable application of LLM is ChatGPT, which presents an amazing ability to interact with humans. OpenAI's ChatGPT and GPT4 are prime examples of the impact that AI can have, and there have been extensive open-source efforts to replicate their success, such as OPT Zhang et al. (2022), BLOOM Scao et al. (2022), PALM Chowdhery et al. (2022), LLaMA Touvron et al. (2023).

Multi-modal large language models have further promoted the development of the vision-language models Radford et al. (2021); Li et al. (2022d); Alayrac et al. (2022); Li et al. (2023); Zhu et al. (2023); Liu et al. (2023a); Chen et al. (2023); Yang et al. (2024; 2025b). CLIP Radford et al. (2021) was introduced to separately extract features from the visual encoder and the text encoder, and combine them using contrastive learning. CLIP supports a variety of downstream tasks, including image retrieval, image classification tasks and especially zero-shot classification tasks. But, it cannot generate detailed captions based on images due to the lack of a text decoder. In contrast, our model primarily addresses the concept drift issue within multi-modal large language models, since an image-grounded text decoder is employed to generate text based on the images. Besides, CLIP requires a large-scale and high-quality WIT dataset to be driven, that contains 37.6 million entity image-text samples with 11.5 million unique images across 108 Wikipedia languages. Whereas, our method is validated under the extended ImageNet-LT, which consists of only 115.8K imbalanced images-text pairs.

Building on CLIP, GLIP Li et al. (2022d) was developed to learn object-level, language-aware, and semantic-rich visual representations, unifying object detection and phrase grounding for pre-training. Different from the contrastive method, Flamingo Alayrac et al. (2022) aligned a pre-trained vision encoder and language model using gated cross-attention, demonstrating impressive few-shot learning capabilities. BLIP2 Li et al. (2023) was subsequently introduced, and it employed a Flan-T5 Chung et al. along with a Q-Former to effectively align visual features with the language model. MiniGPT4 Zhu et al. (2023), the most recent development in the field is the PaLM-E model, which features 562 billion parameters and is designed to integrate real-world continuous sensor modalities into an LLM, thereby establishing a connection between real-world perceptions and human languages. Based on Visual Fundamental Models like BLIP mentioned above, Visual ChatGPT adopts ChatGPT as the central component for interacting with users. It integrates multiple visual foundation models and utilizes prompt engineering, also known as Prompt Manager, to instruct ChatGPT about the usage, input-output format, and capabilities of each foundation model. This enables ChatGPT to determine how to invoke these models to fulfill the user's requirements. Besides, GPT-4V(ision) OpenAI (2023) and GPT-4O(mni) have recently shown unprecedented ability in understanding and processing an arbitrary mix of input images and texts.

### A.1.2 LONG-TAILED OPEN WORLD

In vision tasks, significant efforts have been devoted to mitigating the challenges posed by the long-tailed open world. Two prominent research directions have emerged: long-tailed classification under open-world settings, exemplified by approaches like OLTR++ Liu et al. (2019; 2022b), LUNA Cai et al. (2022), DALC Wang et al. (2023), Open-sampling Wei et al. (2022) and TLC Li et al. (2022a), and OOD detection in long-tailed recognition, as seen in methods such as PASCL Wang et al. (2022), EAT Wei et al. (2024). OLTR++ Liu et al. (2019; 2022b) proposed an ensemble algorithm, consisting of dynamic meta-embedding to improve the recognition of tail categories and active learning for open categories detection. LUNA Cai et al. (2022) presented a distribution-sensitive loss to weigh more on the tail classes and a local-density-based metric to measure the novelty of OOD samples. DALC Wang et al. (2023) designed an active distribution optimization algorithm for clustering, querying and classification to balance the classification bias. Open-sampling Wei et al. (2022) rebalances class priors by sampling labels from a complementary distribution for each open-set

instance, mitigating class imbalance. TLC Li et al. (2022a) utilizes the Dempster-Shafer Evidence Theory in a multi-expert framework for uncertainty estimation of tail and OOD samples. PASCL Wang et al. (2022) applied supervised contrastive learning to explicitly boost the model to distinguish between tail-class in-distribution samples and OOD samples. EAT Wei et al. (2024) introduces abstention classes for clear decision boundaries and augmenting tail classes with context-rich OOD data to focus on discriminative features. MCM Ming et al. (2022) pioneers the integration of vision language models into OOD detection, enabling zero-shot OOD by aligning visual features with text concepts through a proposed maximum concept matching approach.

In addition, more and more VL methods have gained attention in the long-tail domain, such as LPT DONG et al. (2023), BALLAD Ma et al. (2021), Decoder Wang et al. (2024), VL-LTR Tian et al. (2022) and LIFT Shi et al. (2024). However, most of them pay attention to the fine-tuning of the vision language model under long-tailed scenarios. They directly use the pre-trained CLIP model, which is pre-trained using the high-quality and large-scale WIT dataset. In contrast, we are more concerned about the impact of long-tail open data on the whole model training from pre-training onwards, including pre-training and fine-tuning.

Additionally, in the domain of the language model, Kandpal et al. (2023) corroborates that large language models (LLMs) also struggle to learn long-tailed knowledge. While larger models are better at absorbing long-tailed knowledge, they estimate that current models must be scaled by many orders of magnitude to reach competitive performance. Besides, Raunak et al. alleviates the long-tail problem in neural machine translation by quantifying token classification and sequence generation, and introduces an anti-focus loss that incorporates beam search inductive biases to better adapt model training to conditional text generation.

### A.1.3 CONCEPT DRIFT

In the review Lu et al. (2019), the algorithms related to concept drift are categorized into three groups: error rate-based, data distribution-based and multiple hypothesis-based. Our proposed algorithm belongs to the distribution-based concept drift detection and adaptation method. Distribution-based concept drift algorithms not only accurately detect drift through explicit distributions but also analyze the drift to identify its happening timing, location, and severity.

Besides, RBM-IM Korycki & Krawczyk proposes a novel trainable concept drift detector based on Restricted Boltzmann Machine, to solve the concept drift in multi-class imbalanced data streams. Meanwhile, DDG-DA Li et al. initially trains a predictor to estimate future data distribution with concept drift, utilizes this information to create training samples, and subsequently trains models on the generated data. Furthermore, CALMID Liu et al. (2021) proposes a comprehensive active learning method for multiclass imbalanced streaming data with concept drift, including an ensemble classifier, a drift detector, and a variable threshold uncertainty strategy. Subsequently, DES-ICD Jiao et al. (2024) is a dynamic ensemble selection method for imbalanced data streams with concept drift. It considers the local performances of base classifiers and addresses class imbalance using a novel synthetic minority oversampling technique. Moreover, GOOD Gui et al. (2022) develops a graph OOD benchmark, which explicitly distinguishes between covariate and concept shifts and designs data splits that accurately capture these different shifts. Beyond that, ResilientCL Yang et al. (2025a) introduces a causal framework that integrates concept drift adaptation with structural causal modeling. By decoupling spurious correlations via causal graphs and enforcing counterfactual invariance, it addresses distributional biases in streaming training data. Besides, Liu et al. (2022a; 2023b; 2024) propose a multi-view uncertainty framework that addresses concept drift across heterogeneous data streams through set-valued prediction generation, effectively consolidating probabilistic outputs into deterministic categorical representations.

**Remark A.1. Differences: Concept Drift vs. Data Drift (Covariate Drift)** *Data drift entails changes solely in the distribution of inputs $P(x)$, while concept drift involves alterations in both input and output distributions, i.e., $P(x)$ and $P(y)$, leading to changes in the decision boundary. Furthermore, data drift predominantly stems from internal factors like data collection and processing, whereas concept drift typically arises from external factors, reflecting real-world changes.*

### A.1.4 Hyperspherical Distribution Modelling

The Bayesian estimation of the vMF mixture model with variational inference is addressed in Taghia et al.. The learning task in VI consists of the optimization of the variational posterior distribution. Besides, a deep metric learning model for image classification and retrieval is presented in Zhe et al., which utilizes the vMF distribution to define the loss function and introduces an effective alternative learning algorithm by updating class centers. The model captures global information in the embedding space and approximates the class distribution during training, leading to improved performance in image tasks. Kobayashi extends the vMF distribution to regularize the intra-class feature distribution for imbalanced, small-scale and noisy data. Yang et al. (2023) focus on using hyperspherical embedding to alleviate the crowding problem arisen by the imbalanced data. Ming et al. (2023) utilizes hyperspherical embeddings for OOD detection in representation learning, consisting of two losses, a dispersion loss to increase angular distances between different class prototypes, and a compactness loss to ensure samples are closer to their respective class prototypes. Besides, H-SRDC Tang et al. enhances intra-class compactness by combining target data clustering with a domain-shared classifier and cluster centroid learning, enhancing deep clustering by minimizing Kullback-Leibler divergence between network predictions and an auxiliary distribution.

## A.2 The T-distributed Distribution on Hypersphere

### A.2.1 Directional statistics

Directional statistics primarily focus on the distribution of eigenvector angles, while neglecting the impact of eigenvector module lengths. Given the unit feature vector $X_{ij} \in \mathbb{S}^{d-1}$, where $\mathbb{S}^{d-1} = \{x \in \mathbb{R}^d : ||x||_2 = 1\}$ denotes the $(d-1)$-dimensional hyperspherical set. A key idea in directional distribution is the tangent-normal decomposition. Any unit vector $x$ can be decomposed as:

$$x = t\mu + (1-t^2)^{\frac{1}{2}}v, t \in [-1, 1], \tag{9}$$

with $v \in \mathbb{S}^{d-2}$ a tangent to $\mathbb{S}^{d-1}$ at $\mu$ Mardia & Jupp (2000); De Cao & Aziz (2020), where $v$ and $t$ are independent and $v$ is uniform on $\mathbb{S}^{d-2}$. Thus, the intersection of $\mathbb{S}^{d-1}$ with the hyperplane through $t\mu$ and normal to $\mu$ is a $(d-2)$-dimensional sphere of radius $\sqrt{1-t^2}$, that $t$ has density as following:

$$p_T(t; d) \propto (1-t^2)^{\frac{d-3}{2}}, t \in [-1, 1]. \tag{10}$$

Therefore, through the marginal density $p_T$ and $p_v$, we can estimate the density of the entire spherical distribution. One prominent instance is the von Mises-Fisher distribution (vMF) Banerjee et al. (2005), which can be interpreted as a probability distribution over the cosine similarity between a unit vector $x$ and a fixed mean direction $\mu$, following the density:

$$p_X(x; \mu, \kappa) \propto \exp(\kappa\mu^T x), \tag{11}$$

where $\kappa \geqslant 0$ denotes the concentration and $\exp$ represents the exponential function. Therefore, combined with the Eq. 9 and Eq. 10, the density of vMF is:

$$p(x) = C_X(\kappa, d)^{-1} \exp(\kappa\mu^T x), \quad x \sim \text{vMF}(\mu, \kappa)$$
$$C_X(\kappa, d) = \frac{(2\pi)^{d/2} I_{d/2-1}(\kappa)}{\kappa^{d/2-1}}, \tag{12}$$

where $I_m$ denotes the modified Bessel function of the first kind at order $m$.

### A.2.2 Derivation of the T-distributed Distribution on Hypersphere

Given the unit feature vector $X_{ij} \in \mathbb{S}^{d-1}$, where $\mathbb{S}^{d-1} = \{x \in \mathbb{R}^d : ||x||_2 = 1\}$ denotes the $(d-1)$-dimensional hyperspherical set. The proposed T-distribution metric on hypersphere (Thp) follows the density:

$$p_X(x) \propto \frac{2}{\kappa(1 - \mu^T x)}, \tag{13}$$

where $x \in \mathbb{S}^{d-1}$, direction $\mu in \mathbb{S}^{d-1}$ and concentration $\kappa \in \mathbb{R}_{\geq 0}$. Let $T$ bet a random variable that denotes the dot-product $t = \mu^T x$, then $T = 2Z - 1$, with $Z \sim \text{Beta}(\alpha, \beta)$, where $\alpha = \frac{d-1}{2}$ and $\frac{d-3}{2}$.

*Proof.* Given Eq. 10, the marginal distribution of the dot-product $t$ is

$$t \propto \frac{2}{\kappa(1-t)}(1-t^2)^{\frac{d-3}{2}}. \tag{14}$$

So, its normalizer is:

$$
\begin{aligned}
N_T(\kappa, d) &= \int_{\mathbb{S}^{d-1}} \frac{2}{\kappa(1-t)}(1-t^2)^{\frac{d-3}{2}}\,\mathrm{d}t \\
&= \int_{-1}^{1} \frac{1}{\kappa(1-t)}(1+t)^{\frac{d-3}{2}}(1-t)^{\frac{d-3}{2}}\,\mathrm{d}t \tag{15} \\
&= \frac{1}{\kappa}\int_{-1}^{1}(1+t)^{\frac{d-3}{2}}(1-t)^{\frac{d-5}{2}}\,\mathrm{d}t.
\end{aligned}
$$

Given the useful integral function:

$$\int (1+x)^a(1-x)^b\,\mathrm{d}x = 2^{a+b+1}B_{\frac{x+1}{2}}(a+1, b+1) + C. \tag{16}$$

So, its normalizer is:

$$
\begin{aligned}
N_T(\kappa, d) &= \frac{1}{\kappa}2^{d-3}(B_1(\frac{d-1}{2}, \frac{d-3}{2}) - B_0(\frac{d-1}{2}, \frac{d-3}{2})) \\
&= \frac{1}{\kappa}2^{d-3}B(\frac{d-1}{2}, \frac{d-3}{2}). \tag{17}
\end{aligned}
$$

The Beta function:

$$B(a,b) = \frac{\Gamma(a)\Gamma(b)}{\Gamma(a+b)}. \tag{18}$$

So, the normalizer is

$$N_T(\kappa, d) = \frac{1}{\kappa}2^{\alpha+\beta-1}\frac{\Gamma(\alpha)\Gamma(\beta)}{\Gamma(\alpha+\beta)}, \tag{19}$$

where, $\alpha = \frac{d-1}{2}$ and $\beta = \frac{d-3}{2}$. It follows that the probability density function of the marginal distribution of the dot product is,

$$
\begin{aligned}
p_T(t; \kappa, d) &= N_T(\kappa, d)^{-1}\frac{2}{\kappa(1-t)}(1-t^2)^{\frac{d-3}{2}} \\
&= N_T(\kappa, d)^{-1}\frac{2}{\kappa}(1+t)^{\frac{d-3}{2}}(1-t)^{\frac{d-5}{2}} \\
&= N_T(\kappa, d)^{-1}\frac{2}{\kappa}(2z)^{\frac{d-1}{2}-1}(2-2z)^{\frac{d-3}{2}-1} \tag{20} \\
&= \frac{2}{\kappa}B(\alpha, \beta)^{-1}z^{\alpha-1}(1-z)^{\beta-1},
\end{aligned}
$$

where, $\alpha = \frac{d-1}{2}$ and $\beta = \frac{d-3}{2}$. $\qquad\square$

Due to the surface area of the hyper-sphere $\mathbb{S}^{d-1}$ is:

$$A_{d-1} = \frac{2\pi^{\frac{d}{2}}}{\Gamma(\frac{d}{2})}. \tag{21}$$

The T-distributed spherical distribution is expressed via the tangent normal decomposition as a joint distribution between $T \sim p_T t; \kappa, d$ and $V \sim \mathcal{U}(\mathbb{S}^{d-2})$. Since $T \perp\!\!\!\perp V$, the Thp normalizer $N_x(p, k)$ is the product of the normalizer of $p_T(t; \kappa, d)$ and the uniform distribution on $\mathbb{S}^{d-2}$ is:

$$
\begin{aligned}
N_X(\kappa, d) &= N_T(\kappa, d) \cdot A_{d-2} \\
&= 2^{\alpha+\beta-1}B(\alpha, \beta)\frac{2\pi^{\beta}}{\kappa\Gamma(\beta)} \tag{22} \\
&= \frac{2^{\alpha+\beta}\pi^{\beta}}{\kappa}\frac{\Gamma(\alpha)}{\Gamma(\alpha+\beta)}.
\end{aligned}
$$

Thus,

$$p_X(x; \mu, \kappa) = N_X(\kappa, d)^{-1}\frac{2}{\kappa(1-\mu^T x)}. \tag{23}$$

A.3    Implementation Details

For our language-guided image tokenizer, we leverage the strengths of both BERT Devlin et al. (2019b) and ViT as our text encoder, text decoder and visual encoder, respectively.

We employ ViT-Bae as our visual encoder, which consists of 12 transformer encoder layers and an FFN intermediate size of 3,072. The input image size is set to $384 \times 384$, with a patch size of $16 \times 16$. The hidden dimensions of the ViT-Base are 768, with 12 attention heads. And, the number of parameters is about 86 million. Besides, we also use ResNeXt-50 to perform ablation experiments. In addition, ResNeXt-50 has 16 residual blocks with 50 layers. Each block has 3 convolutional layers with the kernel size of $3 \times 3$, the stride of 1 and the padding of 1. The batch normalization and max pooling are utilized to connect the convolutional layers. The classification head hidden dimensions are 2,048.

Additionally, BERT as the language model in our vision-language model, has 12 transformer layers with 768 hidden dimensions and 3,078 intermediate dimensions. The number of attention heads is 12, with the input sequence length of 512. It has approximately 110 million parameters.

In terms of the pre-training progress, the hyperparameters are presented in Table 7. We utilize the AdamW optimizer, which is configured with a cosine annealing schedule as the learning policy. The initial learning rate is set to $2 \times 10^{-5}$, and the AdamW optimizer is employed with hyperparameters $\beta = (0.9, 0.98)$. Additionally, we set the weight decay to 0.05 and the dropout rate to 0.1. During the first 1,000 warm-up steps, the learning rate increases to $2 \times 10^{-5}$, and subsequently decays to $10^{-7}$. Unless otherwise specified, the pre-training of our vision language model consists of 800,000 steps, executed on $2 \times 2$ NVIDIA A100 GPUs. And the pre-training experiments are conducted in the manner of different stages, namely gradual drifts with long-tailed data and sudden drifts with OOD data. It is mainly to compare with different methods with the same setup.

Table 7: The training hyperparameters of our vision language model.

| Pre-training | | Fine-tuning | |
|---|---|---|---|
| Training Steps | 400,000 | Training Steps | 18,000 |
| Warmup Steps | 1,000 | Warmup Steps | 0 |
| Optimizer | AdamW | Optimizer | AdamW |
| Learning Rate | 1e-4 | Learning Rate | 2e-5 |
| Learning Rate Decay | Cosine | Learning Rate Decay | Cosine |
| Adam $\beta$ | (0.9, 0.98) | Adam $\beta$ | (0.9, 0.98) |
| Weight Decay | 0.05 | Weight Decay | 0.05 |
| Batch Size | 50 | Batch Size | 400 |

While in the fine-tuning on the downstream task of classification, the initial learning rate is reduced to $10^{-6}$ without the warmup. The visual encoder and text decoder are frozen out of the training. Thus, the batch size can be increased to 400. The fine-tuning consists of 18,000 steps, executed on $2 \times 2$ NVIDIA A100 GPUs. Other training parameters are the same as the pre-training. Besides, under the only fine-tuning settings, the image encoder and the text encoder are frozen with the CLIP pre-trained parameters, while the image-grounded text decoder is trained during the fine-tuning.

When evaluating the performance of our VL model under the long-tailed open world, we use the top-1 accuracy metric on the downstream classification task. In particular, the categories are split into three groups: many-shot (with more than 100 training samples), medium-shot (with 20-100 training samples), and few-shot (with fewer than 20 training samples). The Top-1 accuracies are computed for each group to evaluate the performance of mitigating the bias introduced by the long-tail distribution, respectively. Furthermore, in order to assess the capability of detecting the OOD drift, we employ two metrics: FPR95 which measures the false positive rate of OOD samples when the true positive rate of ID samples reaches 95%, and AUROC providing the area under the receiver operating characteristic curve. Besides, cosine distance is exploited to measure the distances between features and centers in the feature space of the VL model.

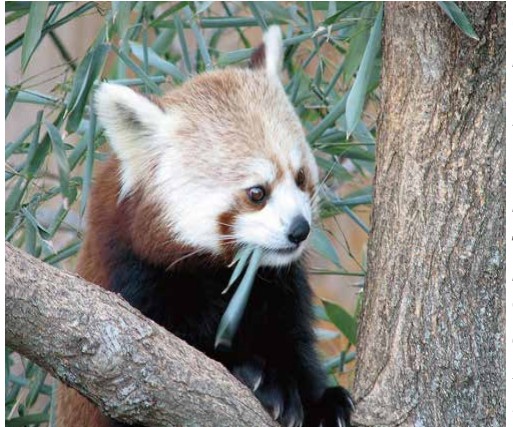

**Caption:** *The image depicts a [mask], also known as a [mask], sitting on a branch of a tree. The [mask] is holding a leaf in its mouth, which suggests that it might be eating or chewing on the plant. This behavior is typical of [mask]s, as they primarily feed on bamboo shoots, leaves, fruits, and insects. In the wild, [mask]s are found in the mountainous regions of southern and southwestern China, Myanmar, and India.*
**Annotation:** *lesser panda, red panda, panda, bear cat, cat bear, Ailurus fulgens*

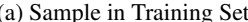

(a) Sample in Training Set

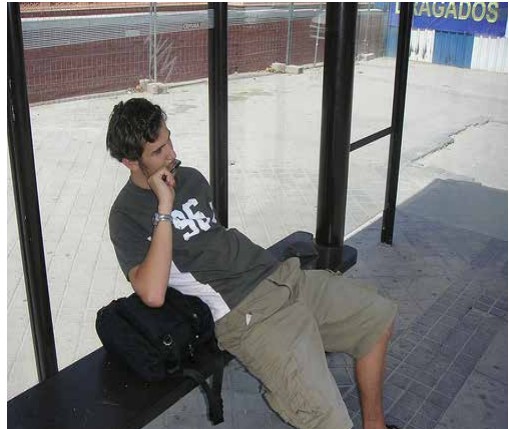

**Caption:** *The picture depicts a young man sitting on a bench, holding a [mask] in his hand. This suggests that he is either playing the [mask] or contemplating playing it. The [mask] is a musical instrument that is commonly associated with blues and folk music, and it can be used to create melodic and rhythmic sounds. The presence of the [mask] in the image adds a musical element to the scene.*

**Annotation:** *harmonica, mouth organ, harp, mouth harp*

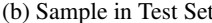

(b) Sample in Test Set

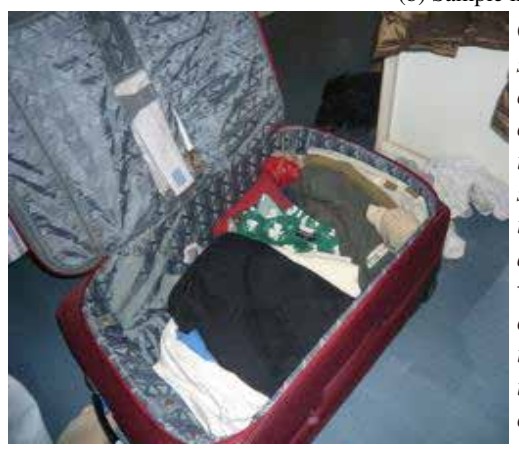

**Caption:** *The main object in the picture is an open suitcase, which is a type of luggage. It is red in color and appears to be medium-sized. The suitcase is located on the floor of a room. The suitcase is partially filled with clothing items, including shirts, pants, and socks. It appears that the suitcase is still in the process of being packed or unpacked, as some items are visible on top of the suitcase while others are spilling out of it. The suitcase is open, allowing easy access to the clothing items inside. Overall, the picture provides a glimpse into the process of preparing for a trip or organizing one's belongings.*

(c) Sample in Open Set

Figure 4: Samples of OpenMMlo in training set, test set and open set.

### A.4 BUILDING MULTI-MODAL LONG-TAILED OOD DATASETS GROUP OPENMMLO

Figure 4 showcases the samples utilized for training and validation in our study. To intuitively verify the impact of long-tail open-world scenarios on multi-modal large language models, we employ classification as our downstream task. When matching images and texts, we strategically mask words that are directly related to category names. This approach ensures the accuracy and reliability of

our experimental results. As depicted in Figure 4, comprehensive descriptions of the image are provided through long-form text, encompassing details such as size, position, color, relationships, and other relevant information about the objects present in the image. This ensures a detailed and information-rich depiction of the visual content. We have publicly released the datasets used for training and validation, as well as the original unmasked datasets.

