# OpenReview forum: "Adapting Multi-modal Large Language Model to Concept Drift From Pre-training Onwards"
_ICLR.cc/2025/Conference — ICLR 2025 Poster_

### Official Review · Reviewer_M73e · 2024-10-29

**Soundness:** 3
**Presentation:** 3
**Contribution:** 3
**Rating:** 8
**Confidence:** 3

**Summary:**

This paper proposes a new unified framework for Multi-modal Large Language Models (MLLMs) to allviate the concept drift. The objective is to enhance MLLMs to enhance classification performance on long-tailed open datasets. The concept drift theory is introduced and extensive experiments demonstrates the method superior performance in downstream tasks. Finally, a multi-modal datasets named OpenMMlo is built for long-tailed open world setting.

**Strengths:**

The paper is well-structured and easy to follow.

The paper conducts extensive experiments, including long-tailed classification and OOD detection.

The paper's results show substantial improvements over other state-of-the-art (SOTA) baselines.

**Weaknesses:**

All downstream tasks only test on vision models, lacking exploration of the overall impact of MLLMs on concept drift. It would be better to provide more experiments, such as long-tailed classification and OOD detection, based on VL models.

It would be better to provide more ablations studies of the T-distributed spherical adapter component, it's not clear that which parts adapt to concept drift and be beneficial to downstream tasks.

**Questions:**

Besides tailed drift and OOD drift, domain shift is also a common problem in open-world environments, especially in the medical field. Can a T-distribution-based drift adapter be robust to domain shift? It would be better to provide experiments to evaluate this on ImageNet-Sketch.

---

> ### Author Response · Authors · 2024-11-21
>
> We sincerely appreciate your insightful and comprehensive review. Your positive feedback serves as a significant encouragement for our team, acknowledging 1) the clear writing, 2) extensive experiments and 3) substantial improvements in experiment results. Additionally, we have meticulously addressed each of your questions in detail, incorporating your feedback into our manuscript revisions.
>
> 1. **Downstream tasks based on VL models.** Thanks for your sincere review. We apologize for the misunderstanding of the experiments. We have provided results of some methods based on VL models in Table.1, such as LPT [1], BALLAD [2], Decoder [3], VL-LTR [4] and LiFT [5]. **We have revised our paper to clarify the misunderstanding in LINE 371 as shown below. Besides, we also provided more results of CLIP under zero-shot, linear probing and fine-tuning to compare our method.**
>
>     > **LINE 371.** It is worth noting that, most vision language models only focus on the impact of tailed drift in the fine-tuning, such as LPT [1], BALLAD [2], Decoder [3], VL-LTR [4] and LiFT [5]. We are the pioneers in revealing the unexplored impacts of concept drift from pre-training onwards.
>
> 2. **Ablations studies of the T-distributed spherical adapter.** Thank you very much for your constructive suggestions. In the ablation experiment, we investigated the efficacy of the T-distributed adapter in addressing the concept drift challenge in vision language models from two perspectives: by examining the impact of removing this module in pre-training and downstream tasks in section 3.3.1, and by conducting an ablation study on the internal concentration parameter kappa of the T-distributed adapter in section 3.3.2. The results verify that the T-distribution adapter's impact is more pronounced in the fine-tuning stage compared to pre-training, since the model is directly involved in specific downstream tasks, and explicit category centers are present in the classifier. In contrast, pre-training primarily focuses on aligning image-text features, where the implicit information of categories is embedded. Additionally, we find $\kappa=16$ yields superior results. We argue that a smaller concentration makes it challenging to effectively mitigate the biases introduced by concept drift in the vision language model.  **We have revised our paper to further explain the ablation experiments in LINE 512.**
>
>
>
> 3. **Domain Shift Results.** Sincerely thanks for your thoughtful and constructive suggestions. Your feedback improves us a lot. **We add experiments on ImageNet-Sketch to evaluate the performance of our method on domain shift in LINE 452.**
>
>
>     |CLIP+ZS [6]|CLIP+LP [6]|CoOp [7]|VPT [8]|DAPT [9]| Ours|
>     |:--:|:--:|:--:|:--:|:--:|:--:|
>     | 46.1 | 36.0 | 47.1 | 47.7 |48.3|50.2|
>
>     > Moreover, we evaluate the generalizability of our method in the domain generalization setting. Experiments are conducted on ImageNet-Sketch [10] with ImageNet [11] as the source dataset, as shown in the Table above. From the experiment results, our method achieves superior performance with an accuracy of 50.2\% on ImageNet-Sketch, attributed to the robustness of the T-distribution-based drift adapter. It further verifies the generalization ability of our model in the open-world.
>
>
> We appreciate your time and thorough review. Your feedback is highly valuable to us, and we welcome further communication from you. If you are satisfied with our response, we would be grateful for your support in enhancing our rating score.
>
> [1] Dong, Bowen, et al. "LPT: Long-tailed prompt tuning for image classification." ICLR 2023.
>
> [2] T. Ma et al., “A Simple Long-Tailed Recognition Baseline via Vision-Language Model,” Nov. 29, 2021, arXiv: arXiv:2111.14745. doi: 10.48550/arXiv.2111.14745.
>
> [3] Y. Wang et al., “Exploring Vision-Language Models for Imbalanced Learning,” IJCV 2024.
>
> [4] C. Tian, W. Wang, X. Zhu, J. Dai, and Y. Qiao, “VL-LTR: Learning Class-wise Visual-Linguistic Representation for Long-Tailed Visual Recognition,”ECCV 2022.
>
> [5] J.-X. Shi, T. Wei, Z. Zhou, J.-J. Shao, X.-Y. Han, and Y.-F. Li, “Long-Tail Learning with Foundation Model: Heavy Fine-Tuning Hurts,” ICML 2024.
>
> [6] A. Radford et al., “Learning Transferable Visual Models From Natural Language Supervision,” ICML 2021.
>
> [7] K. Zhou, J. Yang, C. C. Loy, and Z. Liu, “Learning to Prompt for Vision-Language Models,” IJCV 2022.
>
> [8] M. Jia et al., “Visual Prompt Tuning,” ECCV 2022.
>
> [9] E. Cho, J. Kim, and H. J. Kim, “Distribution-Aware Prompt Tuning for Vision-Language Models,” ICCV, 2023.
>
> [10] Haohan Wang, Songwei Ge, Zachary Lipton, and Eric P Xing. "Learning robust global representations by penalizing local predictive power." NeurIPS, 2019.
>
> [11] Jia Deng, Wei Dong, Richard Socher, Li-Jia Li, Kai Li, and Li Fei-Fei. "Imagenet: A large-scale hierarchical image database." CVPR, 2009.

---

> > ### Comment · Reviewer_M73e · 2024-11-26
> > **Thanks for the response**
> >
> > I have reviewed the authors' rebuttal, and my concerns have been largely addressed. Therefore, I have decided to increase the score.

---

> > > ### Author Response · Authors · 2024-11-27
> > > **Many Thanks for Your Increased Score**
> > >
> > > Thank you very much for your improved rating. Your positive feedback serves as a significant source of encouragement for our team.

---

### Official Review · Reviewer_zKrC · 2024-11-02

**Soundness:** 3
**Presentation:** 2
**Contribution:** 3
**Rating:** 6
**Confidence:** 2

**Summary:**

Tailed drift and OOD drift are known issues in the vision and language community. While such problems have seen extensive studies on vision and language separately, it has not been studied jointly in multimodal learning. This paper attempts to bridge this gap. The authors show that pretraining on an imbalanced dataset worsens inter and intra-class as well as OOD features in terms of separability. These drifts further degrade performance in FT. A T distribution adapter is subsequently adopted to mitigate both drifts. The authors release a new long-tailed dataset pertaining. Extensive experiments on FT classification and OOD detection demonstrate the effectiveness of the proposed method.

**Strengths:**

1. the issue caused by tailed drift and OOD drift (concept shift) is illustrated clearly.
2. The proposed method seems effective across most settings, barring a few in INat which is reasonable.
3. the proposed method is properly theoretically motivated --I checked most math and they seem to be correct.

**Weaknesses:**

1. The manuscript lacks clarity on how their proposed method or model family compares to popular models like CLIP. The related work section of MLLM only discusses the common MLLMs but fails to draw a clear comparison to the proposed method.
2. There lacks motivation on exactly how the proposed method work and why they are designed in this way. The entire section 2 is on defining concept shift as well as T-distribution adapter how ever it doesn't address the architecture. While the authors mention that "drives the training of all three modules", more details are needed here. I also think the authors should clearly state how each loss is calculated, for example, on which labels are the CE loss calculated. This helps reader understand, compare and contrast the proposed method with existing solutions.
3. What advantage would training on OpenMMlo have compared to training on WIT? Are there any other benefits it brings to the table besides supporting the long-tailed study of this paper? -- since naturally we would want balanced dataset in the real world.
4. In the related work section on MLLM, the authors mentions that "However, most of them used the clip pre-trained model, that it is
pre-trained using high-quality and large-scale WIT dataset. And the VL models will also encounter
concept drift caused by data defects in the pre-training, which is the focus of this paper", can you elaborate more on a) would pretraining on WIT mostly already alleviate the said issue and b) what "also encounter concept drift" mean here, odes it refers to pretraining on WIT?
5. How do you define long-tailed data in image-caption pairs? Since no class label is given here how do we define these concepts?
6. The authors mentioned that both gradual and sudden drifts arise from evolving distributions, I wonder if the pre-training experiments are conducted in the manner of different stages?

**Questions:**

See Weakness. My question mainly lies in the clarity of the proposed method as well as dataset, the experiments are alright. While I believe this manuscript has its clear merit, I feel several confounding parts bar readers from a clear understanding and practitioners from adopting the method. I'm inclined to increase my score if the authors can answer my questions and make good efforts to modify their manuscript.

**Details Of Ethics Concerns:**

No ethic concerns are observed.

---

> ### Author Response · Authors · 2024-11-21
>
> Sincerely thanks for your meaningful and detailed review. Your positive feedback is a huge encouragement to our team, including *1) clear issue presented, 2) motivated method and 3) effective validation*. And, we have provided detailed responses below to address your questions one by one, and made revisions to our manuscript according to your feedback, highlighting these changes in red for your reference in the revised version.
>
>
> 1. **Comparison with CLIP.** Thank you very much for your valuable suggestions. **We have revised our Table 1 to add results of CLIP on ImageNet-LT and iNatualist2018 as shown below, and we have added the discussion about the comparison between our method and CLIP in Section 3.1 as shown below. Besides, we have added a detailed introduction to CLIP in related works.**
>
>     |ImageNet-LT|Many|Medium|Few|All|
>     |:--:|:--:|:--:|:--:|:--:|
>     |CLIP + Zero-Shot      | 82.0 | 70.4 | 69.6 | 70.5 |
>     |CLIP + Linear Probing | 87.3 | 65.1 | 19.0 | 67.4 |
>     |CLIP + Fine Tuning    | 83.0 | 65.0 | 39.9 | 68.5 |
>
>     |iNatualist2018|Many|Medium|Few|All|
>     |:--:|:--:|:--:|:--:|:--:|
>     |CLIP + Zero-Shot      | 9.9  | 5.3  | 4.6  | 5.5  |
>     |CLIP + Linear Probing | 62.4 | 7.1  | 0.1  | 10.0 |
>     |CLIP + Fine Tuning    | 79.4 | 67.6 | 59.1 | 65.4 |
>
>
>     > **LINE 417:** We compare our method with the CLIP [1] under zero-shot, linear probing and fine-tuning, and CLIP results are from the Decoder [2]. Based on the zero-shot results, it is evident that CLIP, even trained on large-scale and high-quality WIT datasets, struggles to address the issue of tailed drift. CLIP only achieves 5.5\% in iNaturalist2018, and the accuracy variance between many-shot and medium-shot scenarios is 11.6\% in ImageNet-LT. Our method significantly outperforms CLIP in dealing with tailed drift, especially in iNaturalist2018. It also indicates that training on a high-quality balanced dataset alone cannot effectively mitigate the bias induced by tailed drift. Furthermore, the results of linear probing and fine-tuning demonstrate that imbalanced datasets can induce pronounced tail drift in MLLMs that damage the model performance. The CLIP accuracy in Few-shot is only 39.9\% under fine-tuning on ImageNet-LT, much lower than the 69.6\% under zero-shot. It further verifies the challenges brought by imbalanced data in the training of MLLMs and the superiority of our method in adapting the MLLM to concept drift from pre-training onwards.
>
>     > **LINE 935:** Related Work. CLIP [1] was introduced to separately extract features from the visual encoder and the text encoder, and combine them using contrastive learning. CLIP supports a variety of downstream tasks, including image retrieval, image classification tasks and especially zero-shot classification tasks. But, it cannot generate detailed captions based on images due to the lack of a text decoder. In contrast, our model primarily addresses the concept drift issue within multi-modal large language models, since an image-grounded text decoder is employed to generate text based on the images. Besides, CLIP requires a large-scale and high-quality WIT dataset to be driven, that contains 37.6 million entity image-text samples with 11.5 million unique images across 108 Wikipedia languages. Whereas, our method is validated under the extended ImageNet-LT, which consists of only 115.8K imbalanced images-text pairs.

---

> > ### Author Response · Authors · 2024-11-21
> >
> > 2. **More descriptions about the architecture and losses.** Sincerely thanks for your kind and helpful suggestions. **We have revised section 2.3 to add a more detailed introduction to our architecture and losses in LINE 277 and LINE 302.** In particular, we introduce our three modules in detail, namely an image encoder, a text encoder and an image-grounded text decoder. And we add descriptions to clearly state how each loss is calculated.
> >
> >     > **LINE 277: Architecture.** The proposed vision language model follows the encoder-decoder mixture architecture of the Blip [3], containing an image encoder, a text encoder and an image-grounded text decoder. …… Additionally, an image-grounded text decoder is employed to produce a textual description corresponding to a provided image. Utilizing the input visual features $x_{img}$ and text features $x_{txt}$, we initially create fused multi-modal representations by merging the image and text feature embeddings. These combined features act as the keys and values within the cross-attention blocks in the image-grounded text decoder. Through conditioning on the already predicted partial sequence $y_{i<j}$, the decoder iteratively forecasts the token at position $j$, effectively producing textual descriptions corresponding across modalities.
> >
> >     > **LINE 302: Loss.** Specifically, given a mini-batch with $N$ image-text feature pairs, we calculate the $N \times N$ Thp similarity of the cross between image and text features. $N$ correct pairs are recognized as positive samples to maximize the Thp similarity, whereas the rest of $N^{2}-N$ are negative samples to minimize the similarity. And we follow the ALBEF [4] to use soft labels from a momentum encoder as training targets to account for the potential positives in the negative pairs.
> >     >
> >     > Additionally, coupled with the T-distributed adapter, language modeling loss is utilized to activate the image-grounding text encoder for generating coherent and detailed captions based on the image, further propelling the training of all three modules. Driven by language modeling loss, the model is trained to optimize a cross-entropy loss with label smoothing, to maximize the likelihood of the generated text in an autoregressive manner.
> >
> > 3. **Advantages of OpenMMlo.** We appreciate and thank you for your kind suggestions. It is really a good question. Naturally, we all would like a balanced dataset to train our model. However, as the parameters of large models continue to expand, the demand for extensive training data also escalates. For instance, GPT-4 is trained on roughly 13 trillion tokens, which is approximately 10 trillion words, and it merely contains text without images. The costs involved in cleaning such a large multi-modal dataset to be balanced are enormous. Thus, our aspiration is for the model to adeptly acclimate to the imbalanced dataset by itself, acquiring abundant knowledge with more and more data but not exhibiting bias. OpenMMlo provides a more realistic training environment for vision language models that validates the potential of them to be trained under realistic big data. In contrast, WIT pays more attention to tasks of image retrieval and image classification.  **We have revised section 2.4 to further elaborate on the advantages of our OpenMMlo in LINE 321 as shown below.**
> >
> >     > **LINE 321.** As the parameters of large models continue to expand, the demand for extensive training data also escalates. However, due to the inherent challenge of obtaining images and related captions, most multi-modal datasets struggle to be balanced in an open world, while cleaning the data requires huge costs. Thus, our aspiration is for the model to adeptly acclimate to the imbalanced dataset by itself, acquiring abundant knowledge with more and more data but not exhibiting bias. In this context, a more realistic training dataset for vision language models is required to validate their potential to be trained under the long-tailed open world. Recognizing the demand for higher-quality multi-modal data with long-tailed distribution in an open world, we developed a group of datasets called Open Multi-modal Long-Tailed OOD Datasets (OpenMMlo).

---

> > > ### Author Response · Authors · 2024-11-21
> > >
> > > 4. **Related Work on CLIP Pre-training.** Thanks much for your elaborate and thoughtful review. We will respond to your questions one by one. a) Pre-training on WIT has not already alleviated the long-tailed drift. Since the zero-shot CLIP only achieves 5.5\% accuracy in iNatualist2018, and the accuracy variance between many-shot and medium-shot scenarios is 11.6\% in ImageNet-LT.  b) We are sorry for the misunderstanding. We do not refer that pretraining on WIT will encounter concept drift. We mainly emphasize that our approach focuses on the impact of tailed drift data on model training from the beginning of pre-training. In contrast, many methods only focus on the fine-tuning under long-tailed scenarios, and directly use the CLIP pre-trained model. **We have revised our paper to clarify the misunderstanding in LINE 935. Thanks for your detailed correction again.**
> > >
> > > 5. **Definition of long-tailed data in image-caption pairs.** We sincerely appreciate and thank you for your thoughtful suggestions. That is indeed a good question. In the vision domain, long-tailed data relates to the categories of images. And in the context of NLP,  the work of [5] demonstrates strong correlational and causal relationships between accuracy and relevant text. Thus, we extend image datasets, such as ImageNet-LT and iNatualist2018, and define data imbalance based on the categories labels of ImageNet-LT and iNaturalist2018. Therefore, concerning captions based on images, we counted the word frequencies and found that their distribution is similar to the image categories distribution, which is imbalanced. **We have revised our paper to further explanation in LINE 336.**
> > >
> > > 6. **Pre-training experiments.** Thanks for your review. Yes, you are right, that the pre-training experiments are conducted in the manner of different stages, namely under gradual drifts with long-tailed data and sudden drifts with OOD data. It is mainly to compare different methods with the same setup.
> > >
> > > Thank you once more for generously dedicating your time to provide a thorough review. Your feedback is tremendously valuable, and we are open to hearing from you at any time. If you find our response satisfactory, we would greatly appreciate your assistance in improving our rating score.
> > >
> > > [1] A. Radford et al., “Learning Transferable Visual Models From Natural Language Supervision,” ICML 2021.
> > >
> > > [2] Y. Wang et al., “Exploring vision-language models for imbalanced learning,” IJCV, 2024.
> > >
> > > [3] J. Li, D. Li, C. Xiong, and S. Hoi, “BLIP: Bootstrapping Language-Image Pre-training for Unified Vision-Language Understanding and Generation,” ICML 2022.
> > >
> > > [4] J. Li, R. Selvaraju, A. Gotmare, S. Joty, C. Xiong, and S. C. H. Hoi, “Align before fuse: Vision and language representation learning with momentum distillation,” NeurIPS 2021.
> > >
> > > [5] N. Kandpal, H. Deng, A. Roberts, E. Wallace, and C. Raffel, “Large Language Models Struggle to Learn Long-Tail Knowledge,” ICML 2023.

---

> ### Comment · Reviewer_zKrC · 2024-11-21
> **thanks for the response**
>
> I've read the authors' response and have increased my score given my concerns have been largely addressed.

---

> > ### Author Response · Authors · 2024-11-21
> > **Many Thanks for Your Increased Score**
> >
> > Many thanks for your increased score. Your positive feedback is a huge encouragement to our team.

---

### Official Review · Reviewer_JLLb · 2024-11-03

**Soundness:** 2
**Presentation:** 2
**Contribution:** 3
**Rating:** 8
**Confidence:** 4

**Summary:**

This paper addresses the challenges faced by Multi-modal Large Language Models (MLLMs) due to concept drift in real-world streaming data, which includes gradual drift from long-tailed data and sudden drift from Out-Of-Distribution (OOD) data. While previous research has focused on these issues within the separate domains of vision or language, their effects on Vision-Language (VL) models remain underexplored. The authors propose a unified framework that adapts concept drift theory to the multi-modal context, enhancing the adaptability of VL models to unpredictable distribution changes. They introduce a T-distribution based drift adapter to mitigate biases from gradual drift and improve identification of sudden changes. Extensive experiments demonstrate that their approach enhances image-text alignment accuracy during pre-training and improves performance on various downstream tasks. Additionally, the authors create a new multi-modal dataset, OpenMMlo, designed for long-tailed open-world scenarios to validate their findings.

**Strengths:**

- *Novel Contribution to Concept Drift in MLLMs*: The paper extends concept drift theory to the multi-modal domain, providing a new perspective on how VL models can adapt to unpredictable changes in data distributions, which is a significant advancement in the field.

- *Effective Mitigation Strategies*: The introduction of a T-distribution based drift adapter effectively addresses biases from gradual drift while distinguishing sudden distribution changes, enhancing the robustness and reliability of VL models under concept drift conditions.

- *Empirical Validation and New Dataset*: The extensive experiments demonstrate the proposed method's effectiveness in improving image-text alignment and performance on downstream tasks. The creation of the OpenMMlo dataset specifically tailored for long-tailed open-world settings adds value to the research community by providing a resource for further studies in this area.

**Weaknesses:**

- The authors raise important points regarding the presence of long-tail and Out-Of-Distribution (OOD) issues during the pre-training phase of Multi-modal Large Language Models (MLLMs). However, it would be beneficial for them to clarify why these issues are particularly relevant in the context of pre-training.

Specifically, they should address the following questions:

Relevance of Long-Tail and OOD Issues in Pre-Training: Why do long-tail and OOD issues emerge during the pre-training phase? How do these issues affect the learning process of the model?

Consideration of Sub-Distribution in Training Data: Should the pre-training process take into account the different sub-distributions present in the training data? What implications does this have for the model's performance and adaptability?

Impact of Long-Tail Issues with Large Datasets: Is the long-tail problem still a concern when the pre-training dataset is large? If so, how does the size of the dataset influence the severity of the long-tail issue, and what strategies can be employed to mitigate its effects?

Addressing these questions would strengthen this paper by providing deeper insights into the challenges faced during pre-training and the rationale behind the proposed solutions.

- The used intra-class compactness and T-distribution are somewhat similar to those used in [a]. It would be beneficial to provide a more detailed discussion of the connections between them.

[a] Towards Uncovering the Intrinsic Data Structures for Unsupervised Domain Adaptation using Structurally Regularized Deep Clustering. TPAMI, 2022.

- In Figure 1a, it is unclear whether the test sets used to evaluate the BL (Balanced) or LT (Long-Tailed) trained VL (Vision-Language) models are the same. Additionally, it is important to know if these test sets are balanced in terms of class distribution. The answers to these questions could significantly impact the empirical evaluation results and the conclusions drawn from the study. I recommend that the authors clarify this aspect, as it is crucial for understanding the validity and reliability of the comparisons made between the two pretraining setups.

- In Figure 1a, it is unclear whether the Out-Of-Distribution (OOD) samples are included in the pre-training of the Vision-Language (VL) model or if they are used solely for evaluation purposes. Additionally, the authors should clarify how OOD drift impacts the pre-training of the VL model.

It would be helpful to understand whether the performance is assessed on both In-Distribution (ID) test samples and OOD samples after incorporating OOD samples into the training process. This clarification is essential for comprehensively evaluating the model's robustness and adaptability in the presence of OOD drift.

- In Figure 1b, it is unclear whether the Vision-Language (VL) model used for evaluation has been fine-tuned on a downstream class-imbalanced dataset. Additionally, the authors should discuss whether fine-tuning on a balanced dataset can mitigate the side effects of pre-training on an imbalanced dataset.

- The writing and overall flow of the article should be further improved. Enhancing clarity and coherence will help convey the authors' ideas more effectively.

- In Fig. 1a, 'LL' in the green rounded rectangle should be "LT"?

- In Abstract, “distribution unpredictable changes” should be "unpredictable distribution changes”.

**Questions:**

See Weaknesses.

---

> ### Author Response · Authors · 2024-11-21
>
> We sincerely appreciate your thoughtful and detailed review. Your positive feedback serves as a great source of encouragement for our team, especially recognizing the *1) novel contribution to concept drift in MLLMs, 2) effective mitigation strategies, 3) extensive experiments and 4) new datasets for long-tailed open-world settings*. Furthermore, we have diligently responded to each of your questions and incorporated your feedback into our manuscript revisions, with these changes highlighted in red for your reference in the revised version.
>
> - **Relevance of Long-Tail and OOD Issues in Pre-Training.** Thanks for your kind suggestions.  As shown in Figure 1 of our manuscript, we conduct a comparison experiment of the VL model trained on the balanced dataset ImageNet and the imbalanced dataset ImageNet-LT, to illustrate the relevance of long-taile and OOD in the pre-training. It is evident that training on the imbalanced dataset leads to a higher degree, indicating worse intra-class compactness brought by the tailed drift. It leads to an overall performance degradation in the pre-training. Beyond that, the undistinguished OOD drift will bias the underlying distribution of the feature space in the VL model, further disturbing the image-text alignment in the pre-training.
>
> -  **Sub-Distribution in Training Data.** Many thanks for your kind review. We indeed take into account the different sub-distributions present in the training data. It is mainly reflected in the robustness and generalization ability of our model. **We have revised our paper and added experiments on ImageNet-Sketch to evaluate the performance of our method on robustness and generalization as shown below.** It verifies the robustness and generalization of our model in dealing with sub-distribution in training data.
>
>     |CLIP+ZS [6]|CLIP+LP [6]|CoOp [7]|VPT [8]|DAPT [9]| Ours|
>     |:--:|:--:|:--:|:--:|:--:|:--:|
>     | 46.1 | 36.0 | 47.1 | 47.7 |48.3|50.2|
>
>
> - **Impact of Long-Tail Issues with Large Datasets.** Thanks for your constructive suggestions. When the training data is large, the large model still cannot avoid the adverse effects of long-tail data on the model. **We have added results of CLIP under zero-shot to explain the question.** Even though CLIP is training in large-scale and high-quality WIT dataset, it only achieves 5.5\% accuracy in iNatualist2018 dataset, further demonstrating that MLLMs still suffer from the tailed drift, despite the training datasets being large.
>
>
> - **Related Work.** Thanks for your kind advice. **We have revised our paper, discussed the related work of H-SRDC [1] and cited this paper in LINE 1028.** H-SRDC [1] enhances intra-class compactness by combining target data clustering with a domain-shared classifier and cluster centroid learning, enhancing deep clustering by minimizing Kullback-Leibler divergence between network predictions and an auxiliary distribution. In contrast, We utilize the light-tailed characteristics of the T-distributed adapter, which effectively counteracts the squeezing of tail categories caused by an overwhelming number of head samples, thereby alleviating the bias induced by tailed concept drift.
>
> - **Pre-training in Figure 1 (a).** Thanks for your detailed suggestions. **We have revised our paper and added more detailed descriptions about the test sets and OOD samples in the caption of Figure 1.** Firstly, we use the same test sets to evaluate the balanced and long-tailed trained VL models to illustrate the impact of tailed drift ti MLLMs. And these test sets are balanced in terms of class distribution. Second, OOD samples are not included in the pre-training of the VL model. We argue that VL models will suffer from OOD drift during inference, so we use OOD data during evaluation. And we find that compared to training on the balanced dataset, the VL model trained under an imbalanced scenario is harder to distinguish between ID samples and OOD samples from the open world due to their similar inter-class separability.
>
> - **Fine-tuning in Figure 1 (b).** Many thanks for your advice. **We have revised our paper and added more detailed descriptions about the test sets and OOD samples in the caption of Figure 1.**. The VL model used for evaluation is fine-tuned on imbalanced datasets.
>
> - **Proofreading.** Thank you very much for your feedback. **We have meticulously proofread the entire paper to enhance its clarity and coherence, involving rectifying any typos and conducting a thorough review of the manuscript to ensure it is error-free.**
>
> Thank you for your continued time and thorough review. Your feedback is immensely valuable to us, and we are always eager to hear from you. If you find our response satisfactory, could you please consider helping us improve our rating score?
>
> [1] Towards Uncovering the Intrinsic Data Structures for Unsupervised Domain Adaptation using Structurally Regularized Deep Clustering. TPAMI, 2022.

---

> ### Comment · Reviewer_JLLb · 2024-11-26
>
> I have reviewed the authors' rebuttal as well as the comments from the other reviewers. I appreciate the thorough discussions, analyses, and experiments presented, which effectively address many of my concerns. The revised paper does adequately incorporate the suggested works for discussion or experiments, which is good. In light of these, I will raise my current rating.

---

> > ### Author Response · Authors · 2024-11-27
> > **Many Thanks for Your Increased Score**
> >
> > We greatly appreciate your higher rating. Your positive feedback is incredibly helpful and a tremendous boost for our team.

---

### Official Review · Reviewer_8prm · 2024-11-05

**Soundness:** 3
**Presentation:** 2
**Contribution:** 3
**Rating:** 6
**Confidence:** 4

**Summary:**

This paper proposed a T-distribution based drift adapter for the vision language model to reduce the potential impact of concept drift in real-world data streams. Meanwhile, a multi-modal dataset OpenMMlo is built in a long tail open world setting.

**Strengths:**

1. This model has good performance on both long-tail and OOD datasets.
2. The motivation of the paper is clear and detailed experiments and pictures are provided as support.

**Weaknesses:**

1. The introduction and evaluation of the image-grounded decoder and text modeling loss are lacking. The author did not introduce and evaluate this module in the methods and experiments. What is the motivation and function of introducing this module?
2. The setup of the experiment is inappropriate. Although OpenMMlo is presented as part of the paper contribution, it is not sufficient to show only the training results of OpenMMlo, which makes comparisons with other methods unfair and meaningless. The authors are encouraged to align the training strategy with the remaining methods, and the current experiments make it impossible to evaluate both the method and the new data set.
3. The resolution used in the paper is 336x336, and the performance of LIVT should be compared with that of the corresponding resolution.
4. Typos in Fig. 1, 'LL' should be 'LT'.

**Questions:**

1. The current experiment, which introduces additional data, larger resolution, and additional modules, makes it impossible to effectively evaluate the effectiveness of the method, which is my main concern in giving a rating.
2. The text decoder used in the fine-tuning stage is trained from the pretrain stage, so how could the method only fine-tuning using the pretrained CLIP model?

---

> ### Author Response · Authors · 2024-11-21
>
> Thank you very much for your review, especially for summarizing the strengths of our work, including *1) motivation 2) detailed experiments and 3) good performance*. In response to your concerns, we have provided a detailed explanation below and revised our manuscript accordingly, with all changes highlighted in red in the revised version.
>
> 1. **Weaknesses 1: image-grounded decoder and text modeling loss.** Thanks for your kind suggestions. The motivation of the image-grounded decoder is applying text modeling to train the vision language model driven by the text modeling loss, in addition to aligning image-text by contrastive learning. They are introduced in Blip [1] to implement both vision-language understanding and generation tasks, and we are following them to generate text based on images. **We added the following descriptions in the Section 2.3 of the revised manuscript to clarify the misunderstanding, marked in red.** Thanks for your advice again.
>
>     > **LINE 292:** Additionally, an image-grounded text decoder is employed to produce a textual description corresponding to a provided image. Utilizing the input visual features $x_{img}$ and text features $x_{txt}$, we initially create fused multi-modal representations by merging the image and text feature embeddings. These combined features act as the keys and values within the cross-attention blocks in the image-grounded text decoder. Through conditioning on the already predicted partial sequence $y_{i<j}$, the decoder iteratively forecasts the token at position $j$, effectively producing textual descriptions corresponding across modalities.
>
>     > **LINE 307:** Additionally, coupled with the T-distributed adapter, language modeling loss is utilized to activate the image-grounding text encoder for generating coherent and detailed captions based on the image, further propelling the training of all three modules. Driven by language modeling loss, the model is trained to optimize a cross-entropy loss with label smoothing, to maximize the likelihood of the generated text in an autoregressive manner.
>
>
> 2. **Weaknesses 2: The setup of the experiment.** Thanks very much for your detailed and thoughtful review. We found it is similar to the Question 1, so we answered both questions together here. We will answer your questions from the following three aspects:
>
>    -  Additional data. We sincerely apologize for the misunderstanding caused to you. Despite our created OpenMMlo extends ImageNet-LT, iNatualist2018 and Places-LT with elaborate descriptions, we did not combine three datasets to pre-train the model. When we conduct long-tail evaluation, we still use the same dataset as other methods without additional data. **We have revised our manuscript to clarify the misunderstanding in LINE 358 of Section 3.1, marked in red.**
>
>     - Larger resolution. Thanks for your kind suggestions. Your feedback is essential to our improvement process. **We have revised Table 1 in our revised manuscript, which added the results with the resolution of $224\times 224$ as shown below.** Even at low resolution, our method still outperforms other methods when training from scratch or only fine-tuning. Besides, we follow the LiVT [3] to present the results with a large resolution, showing our model can also benefit from a larger image input.
>
>         ||Many|Medium|Few|All|
>         |:--:|:--:|:--:|:--:|:--:|
>         |*Training from Scratch*
>         |ImageNet-LT       | 76.4 | 66.2 | 48.9 | 68.0 |
>         |iNaturalist 2018  | 82.5 | 79.8 | 77.1 | 78.9 |
>         |*Only Fine-tuning*
>         |ImageNet-LT       | 79.5 | 76.5 | 74.1 | 77.2 |
>         |iNaturalist 2018  | 83.5 | 82.2 | 80.7 | 81.7 |
>
>     - Additional modules. Thanks very much for your suggestions, and we fully understand your concerns. However, the additional module of the T-distributed adapter does not have learnable parameters, that we only tested kappa as a learnable parameter in the ablation experiment. Therefore, the improvement of our methods is not attributed to the additional parameters from extra modules.

---

> > ### Author Response · Authors · 2024-11-21
> >
> > 3. **Weaknesses 3: The resolution.** Thanks very much for your advice. **We have revised Table 1 in our revised manuscript, which added the results with the resolution of $224\times 224$ as shown in Answer 2.** Even at low resolution, our method still outperforms other methods when training from scratch and only fine-tuning.
> >
> >
> > 4. **Weaknesses 4: Typos.** Thanks very much for your corrections. **We have corrected this typo and double-checked the entire manuscript to make sure there are no further typos.**
> >
> >
> > 5. **Question 1: The setup of the experiment.** Thanks for your comments. We found it is similar to the Weaknesses 2, so we answered both questions together in weaknesses 2. Thanks very much!
> >
> >
> > 6. **Question 2: Fine-tuning.** Thanks for your detailed review. We are sorry for the misunderstanding of only fine-tuning. As we claimed in Section 3.1, *we apply the same setup as LIFT[2], i.e., using the pre-trained model of the clip and only fine-tuning.* Only fine-tuning means that the method does not pre-train the model on the long-tail dataset, while directly using the parameters of CLIP pre-trained on high-quality and large-scale WIT dataset. And they only fine-tune the model on long-tailed datasets. Under the only fine-tuning settings, the image encoder and the text encoder are frozen with the CLIP pre-trained parameters, while the image-grounded text decoder is trained during the fine-tuning. **We have clarified this misunderstanding in LINE 369 and LINE 1166 of our revised manuscripts denoted in red.**
> >
> >
> >
> > Thanks again for your valuable time and careful review. Your feedback is incredibly helpful, and we’d love to hear from you anytime. If you find our response satisfactory, would you kindly help us in enhancing our rating score?
> >
> > [1] J. Li, D. Li, C. Xiong, and S. Hoi, “BLIP: Bootstrapping Language-Image Pre-training for Unified Vision-Language Understanding and Generation,” ICML 2022.
> >
> > [2] J.-X. Shi, T. Wei, Z. Zhou, J.-J. Shao, X.-Y. Han, and Y.-F. Li, “Long-Tail Learning with Foundation Model: Heavy Fine-Tuning Hurts,” ICML 2024.
> >
> > [3] Z. Xu, R. Liu, S. Yang, Z. Chai, and C. Yuan, “Learning Imbalanced Data With Vision Transformers,” CVPR, 2023.

---

> > ### Comment · Reviewer_8prm · 2024-11-25
> > **Thanks for the response**
> >
> > Thank you for your patient reply and some of my questions have been solved.
> > However, based on the current rebuttal and experimental results, I still cannot properly evaluate the effectiveness of OpenMMlo and the impact of different modules.
> > I will raise my score, but I think there is a lot that could be improved in this paper.

---

> > > ### Author Response · Authors · 2024-11-25
> > > **Many Thanks for Your Increased Score**
> > >
> > > Thank you very much for your response and the increased score. Your positive feedback really encouraged us. Moreover, to address your concerns, we provide further explanations about OpenMMlo: Currently, the majority of research on the tailed drift of VL models only uses straightforward and simple captions, such as "a photo of [category]." Since they are based on the CLIP architecture, such as LiFT [1], Decoder [2] and VL-LTR [3]. However, we argue that short and simplistic captions may not adequately represent the imbalanced nature of realistic textual data encountered during MLLM training. Thus, we create OpenMMlo, an imbalanced multi-modal dataset with longer and more complex texts. It is a more challenging task and more meaningful for the VL models in the real long-tailed open world. Additionally, we will further refine our paper to enhance its presentation and clarity.
> > >
> > >
> > > [1] J.-X. Shi, T. Wei, Z. Zhou, J.-J. Shao, X.-Y. Han, and Y.-F. Li, “Long-Tail Learning with Foundation Model: Heavy Fine-Tuning Hurts,” ICML,2024
> > >
> > > [2] Y. Wang et al., “Exploring Vision-Language Models for Imbalanced Learning,” IJCV, 2024.
> > >
> > > [3] C. Tian, W. Wang, X. Zhu, J. Dai, and Y. Qiao, “VL-LTR: Learning Class-wise Visual-Linguistic Representation for Long-Tailed Visual Recognition,” ECCV 2022.

---

### Author Response · Authors · 2024-11-25
**Kind reminder of the discussion**

We sincerely thank all reviewers and ACs for your constructive and detailed reviews. We have answered the reviewers' questions one by one, and revised our paper according to the suggestions, denoted in red in our revised paper. we really appreciate that Reviewer zKrC has read our response and increased the score. We would like to know if our response has addressed your concerns and questions. If you have any further concerns or suggestions for the paper or our rebuttal, please let us know. We would be happy to engage in further discussion and manuscript improvement.

---

### Meta-Review · Area_Chair_A5Lp · 2024-12-18

**Metareview:**

The paper addresses the problem of concept drift for MLLMs with a new adapter-based approach. The paper received four reviews with 2x accept and 2x borderline accept ratings. The reviews are generally positive: the reviewers appreciated the idea and the results; they also found the paper well-motivated and the writing easy to follow. On the negative side, the reviewers mainly criticized that the paper lacks clarity in several places. The rebuttal had properly addressed the reviewers' questions.

**Additional Comments On Reviewer Discussion:**

The reviewers pointed out several places where the statements and explanations are unclear and thus need more clarifications. The rebuttal has properly addressed these. The authors are strongly suggested to include these discussions in the paper.

---

### Decision · Program_Chairs · 2025-01-22

Accept (Poster)